# Asynchronous Decentralized SGD for Non-Convex Optimization via a Block-Coordinate Descent Lens

## Abstract

Decentralized optimization has become vital for leveraging distributed data without central control, enhancing scalability and privacy. However, practical deployments face fundamental challenges due to heterogeneous computation speeds, unpredictable communication delays, and diverse local data distributions. This paper introduces a refined model of Asynchronous Decentralized Stochastic Gradient Descent (ADSGD) under practical assumptions of bounded computation and communication times. To analyze its convergence for non-convex objectives, we first study Asynchronous Stochastic Block Coordinate Descent (ASBCD) as a theoretical tool, and employ a *double-step-size technique* to handle the interplay between stochasticity and asynchrony. This approach allows us to establish convergence of ADSGD under *computation-delay-independent* step sizes, without assuming bounded data heterogeneity. Empirical results show that ADSGD is practically robust even under extreme data heterogeneity and can be multiple times faster than existing methods in wall-clock convergence. With its simplicity, efficiency in memory and communication, and resilience to delays, ADSGD is well-suited for real-world decentralized learning tasks.

## 1 Introduction

In the era of deep learning, especially with the dominance of Large Language Models, training datasets are getting larger and sometimes are spatially distributed. Consequently, centralized training is often not desired and even impossible due to either memory constraints or the decentralized nature of data. Decentralized optimization (DO), therefore, becomes a perfect remedy (Tang et al., 2023). It aims to minimize the sum of local objective functions, i.e.,

$$\min_{x \in \mathbb{R}^d} f(x) = \sum_{i=1}^{n} f_i(x), \tag{1}$$

where $n$ is the number of agents and $d$ is the dimension of the problem. The optimization process is decentralized in that each agent only has access to the local objective function $f_i$. In deep learning, one typical form of $f_i$ is $f_i(x) \triangleq \mathbb{E}_{\xi \sim \mathcal{D}_i} F_i(x; \xi)$, where $\mathcal{D}_i$ represents the local data distribution of agent $i$, and $F_i$ is the loss function.

Most decentralized methods (Pu et al., 2020; Nedic et al., 2017) use synchronous updates, suffering from stragglers in heterogeneous systems. Asynchronous approaches avoid this bottleneck and often perform better in practice (Samarakoon et al., 2019). However, existing asynchronous methods (Lian et al., 2018; Niwa et al., 2021; Bornstein et al., 2022; Koloskova et al., 2020) typically require either partial synchronization or activation assumptions (e.g., independent sampling with fixed probabilities). For example, ADPSGD (Lian et al., 2018) imposes strict synchronization requirements: (1) agents must maintain identical update frequencies, and (2) neighbor synchronization is mandatory during updates. These constraints create significant waiting times during execution.

In this work, we propose Asynchronous Decentralized SGD (ADSGD) method and perform analysis under the assumptions that require only bounded computation/communication delays (Section 3.1). The analysis connects ADSGD to Asynchronous Stochastic Block Coordinate Descent (ASBCD), providing new convergence guarantees for non-convex objectives.

### 1.1 RELATED WORK

This section reviews the literature on Asynchronous Block Coordinate Descent (ABCD) and asynchronous decentralized optimization algorithms. Note that most of the asynchronous algorithms make probabilistic assumptions regarding update patterns (Lian et al., 2018; Bornstein et al., 2022; Koloskova et al., 2022; Liu & Wright, 2015; Peng et al., 2016; Leblond et al., 2017). While these assumptions simplify theoretical analysis, they may not accurately approximate real-world scenarios. Here, we focus exclusively on algorithms that align with the same asynchrony assumptions as those adopted in this work.

**ABCD methods.** Readers might refer to (Sun et al., 2017) for a slightly outdated review. More recently, several works (Kazemi & Wang, 2019; Ubl & Hale, 2022; Zhou et al., 2018) investigated the proximal block coordinate descent method. The work in (Ubl & Hale, 2022) considers the convex setting. While (Zhou et al., 2018) extends to the non-convex case, additional assumptions such as the Luo-Tseng error bound condition (Tseng, 1991) are required. Note that these studies do not provide a convergence rate. In (Sun et al., 2017), a convergence rate of $o(\frac{1}{\sqrt{k}})$ is established for non-convex problems. The paper (Kazemi & Wang, 2019) proposes an accelerated algorithm and achieves similar results as in (Sun et al., 2017). However, they have not considered the stochastic gradient setting.

**Asynchronous decentralized optimization methods.** Most asynchronous decentralized optimization methods study deterministic gradients, primarily using tracking-based approaches. For example, the works in (Cannelli et al., 2020) and (Zhang & You, 2019) both achieve linear convergence, where APPG (Zhang & You, 2019) assumed PŁ-condition and (Cannelli et al., 2020) used the Luo-Tseng error bound condition (Tseng, 1991). The work in (Tian et al., 2020) achieves sublinear convergence for general non-convex functions, but suffers from: (1) step sizes scaling as $\mathcal{O}(\underline{w}^{(2n-1)B+nD})$[1], $B, D$ are the bounds of computation and communication delays, respectively. (2) heavy memory/communication overhead due to gradient tracking.

While the paper (Wu et al., 2023) proves delay-agnostic convergence for asynchronous DGD with exact gradients under strongly convex objectives, extending the result to non-convex problems with stochastic gradients is non-trivial. Specifically, the max-block pseudo-contractive analysis fails for stochastic gradients due to non-commutativity of max and expectation operations.

Under the adopted asynchrony assumptions of bounded computation/communication delays, existing methods using stochastic gradients either focus on strongly convex objectives (Spiridonoff et al., 2020) or impose strong constraints for non-convex cases. For example, under a simple case - a 3-agent fully-connected network with no delays and 1-smooth loss - the tracking-based methods in (Zhu et al., 2023) and (Kungurtsev et al., 2023) theoretically require step sizes below $2.2 \times 10^{-27}$ and $3.5 \times 10^{-54}$. In contrast, the theoretical step size for ADSGD has a clear and simple dependency on $D$ and $K$ (total iteration number) only. Moreover, tracking-based methods require over three times as much memory in practice and double the communication budget compared with ADSGD. Readers can refer to Appendix A.1 for a detailed comparison.

### 1.2 MAIN CONTRIBUTIONS

- We introduce **ADSGD**, an asynchronous algorithm that provably converges for non-convex objectives with stochastic gradients under *bounded computation and communication delays*, using step sizes that are independent of computation delays. Compared to existing methods, ADSGD reduces per-iteration communication overhead by 50% and memory usage by 70%. Notably, its convergence guarantee does not rely on the commonly assumed bounded data heterogeneity condition.

- We analyze ADSGD through the lens of block-coordinate descent. As a first step, we generalize asynchronous block coordinate descent (ABCD) to the stochastic setting, obtaining ASBCD, and establish its convergence under non-convex objectives. However, this convergence analysis does not directly extend to ADSGD: a naive attempt to treat ADSGD within the ASBCD framework yields a step-size–dependent Lipschitz constant, which in turn leads to divergence. To overcome this obstacle, we develop a *double-step-size tech-*

---

[1] $\underline{w}$ is the lower bound of the weights in the weight matrix

*nique*, which decouples the Lipschitz dependency and enables us to establish the convergence of ADSGD.

- We demonstrate empirically that ADSGD converges faster than all baselines (including synchronous and asynchronous methods) regardless of stragglers, showing that the method is delay-resilient, communication/memory-efficient, and simple to implement, making it well-suited for practical deployment.

## 2 THE ALGORITHMS

This section details the ASBCD and ADSGD algorithms.

### 2.1 ASBCD

Consider the optimization problem

$$\min_{\mathbf{x} \in \mathbb{R}^{d'}} \mathbf{f}(\mathbf{x}), \tag{2}$$

where $\mathbf{x} = (x_1^T, ..., x_n^T)^T, x_i \in \mathbb{R}^{d_i}$, and $\sum_{i=1}^{n} d_i = d'$.

In ASBCD, each block stores part of the global model $x_i$ and a buffer $\mathcal{B}_i$, containing all blocks of the global model $\{x_{ij}\}_{j \in [n]}$. Specifically, $x_i$ is the current local iterate of block $i$ and $x_{ij}$ records the most recent $x_j$ it received from block $j \in \mathcal{N}_i$. As in Algorithm 1, each block keeps estimating $\nabla_i \mathbf{f}(\cdot)$ with the global model in the buffer. Once the gradient estimation is available, block $i$ updates as follows:

$$x_i \leftarrow x_i - \alpha g_i^{\mathbf{f}}(\mathbf{x}_i), \tag{3}$$

where $\mathbf{x}_i = (x_{i1}^T, ..., x_{in}^T)^T$ and $g_i^{\mathbf{f}}(\cdot)$ is a stochastic estimator of $\nabla_i \mathbf{f}(\cdot)$. Then block $i$ sends the updated block to all other blocks and repeats.

For a clear mathematical representation, we introduce the virtual iteration index $k \in \mathbb{N}_0$, which is only for analysis and need not be known by any block. The index $k$ is increased by 1 whenever some block is updated. The updating rule can be written as

$$x_i^{k+1} = \begin{cases} x_i^k - \alpha g_i^{\mathbf{f}}(\hat{\mathbf{x}}^k), & i = i_k, \\ x_i^k, & \text{otherwise,} \end{cases} \tag{4}$$

where $i_k$ is the active block at step $k$ and $\hat{\mathbf{x}}^k$ is the global model held by block $i_k$ at iteration $k$. Note that the read of the global model is done prior to gradient calculation, hence $\hat{\mathbf{x}}^k$ is the available global model to $i_k$ when it begins the gradient estimation before iteration $k$. For instance, suppose a block starts gradient estimation at iteration 2 and takes 3 iterations to finish. Then $\hat{\mathbf{x}}^5$ is the global model available to $i_5$ at iteration 2. Specifically, $\hat{\mathbf{x}}^k = ((x_1^{s_{i_k 1}^k})^T, ..., (x_n^{s_{i_k n}^k})^T)^T$ and $s_{ij}^k \le k$ is the largest iteration index (but smaller than or equal to $k$) of the most recent version of $x_j$ available to block $i$ when it starts its last gradient estimation prior to iteration $k$. In other words, $s_{ij}^k$ is the minimal value of the current iteration index $k$ and the iteration when block $j$ conducts its next update. Note that $s_{ij}^k$ degenerates to a simpler term under certain circumstances, e.g. $s_{ii}^k = k$.

Fig. 1a and 1b provide examples of $s_{ij}^k$. In both figures, block $i_9$ starts calculating gradient at iteration 2 and 4 and finishes at iteration 4 and 9. We first focus on $s_{i_9 1}^9$, which is the largest iteration index of the most recent $x_1$ available when $i_9$ starts its last gradient estimation at iteration 4. It is shown in both figures that $x_1$ is available at iteration 2. However, block 1 updates at iteration 5 and 3 in Fig. 1a and 1b, respectively. Thus $s_{i_9 1}^9 = 5$ for Fig. 1a and $s_{i_9 1}^9 = 3$ for Fig. 1b. Next, consider $s_{i_4 1}^4$, which is the largest iteration index of the most recent $x_1$ available at iteration 2. Now, $s_{i_4 1}^4 = \min\{5, 4\} = 4$ for Fig. 1a because block 1 remains unchanged until iteration 5. Similarly, $s_{i_4 1}^4 = \min\{3, 4\} = 3$ for Fig. 1b.

**Remark 2.1.** $s_{ij}^k$ here captures both computation delay and communication delay. The term $k - s_{ij}^k$ will be large when either $i$ is slow at computing (while $j$ is fast) or the communication link from $j$ to $i$ is slow. If $\{i_k\} = [n]$ (all blocks update together) and $s_{ij} = k$ for all $i, j, k$, equation 4 degenerates to standard stochastic gradient descent.

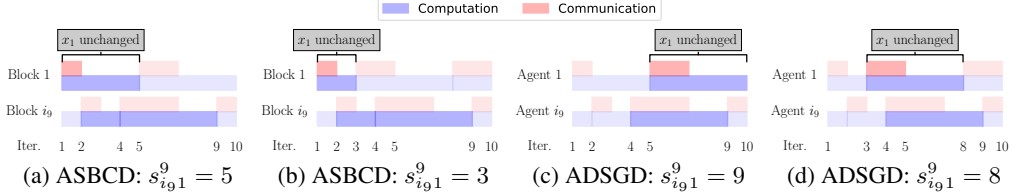

(a) ASBCD: $s_{i_9 1}^9 = 5$  (b) ASBCD: $s_{i_9 1}^9 = 3$  (c) ADSGD: $s_{i_9 1}^9 = 9$  (d) ADSGD: $s_{i_9 1}^9 = 8$

Figure 1: Schematics of $s_{ij}^k$ for ASBCD (left) and ADSGD (right), with the computation and communication of primary focus highlighted in a darker shade.

---

**Algorithm 1** ASBCD
---

1: **Initialization:** All blocks agree on $\alpha > 0$.
2: Each block chooses $x_i$, creates a local buffer $\mathcal{B}_i$, shares $x_i$, and calculates $g_i^{\mathbf{f}}(\mathbf{x}_i)$.
3: **All Blocks Do In Parallel:**
4: **while** termination criterion not met **do**
5:    **repeat**
6:       Keep receiving $x_j$ from other blocks.
7:       Let $x_{ij} = x_j$ and store $x_{ij}$ in $\mathcal{B}_i$
8:    **until** $g_i^{\mathbf{f}}(\mathbf{x}_i)$ is available.
9:    Update $x_i$ according to equation 3.
10:   Send $x_i$ to every other block.
11:   Calculate $g_i^{\mathbf{f}}(\mathbf{x}_i)$.
12: **end while**

---

## 2.2 ADSGD

ADSGD aims to solve equation 1 over a network of $n$ agents described by an undirected, connected graph $\mathcal{G} = (\mathcal{V}, \mathcal{E})$, where $\mathcal{V} = \{1, \dots, n\}$ is the vertex set and $\mathcal{E} \subseteq \mathcal{V} \times \mathcal{V}$ is the edge set. In the network, each agent $i$ observes a local cost function $f_i : \mathbb{R}^d \to \mathbb{R}$ and can only interact with its neighbors in $\mathcal{N}_i = \{j : \{i, j\} \in \mathcal{E}\}$.

In asynchronous DSGD, each node behaves similarly as in ASBCD. Every node $i \in \mathcal{V}$ holds the model of itself $x_i$ and a buffer $\mathcal{B}_i$, which contains the models of its neighbors $\{x_{ij}\}_{j \in \mathcal{N}_i}$. Like before, $x_i$ is the current local iterate of node $i$, and $x_{ij}$ records the most recent $x_j$ it received from node $j \in \mathcal{N}_i$. Each agent keeps estimating $\nabla f_i(x_i)$ and updates as follows once finished,

$$x_i \leftarrow w_{ii} x_i + \sum_{j \in \mathcal{N}_i} w_{ij} x_{ij} - \alpha g^{f_i}(x_i), \tag{5}$$

where $g^{f_i}(x_i)$ is a stochastic estimator of $\nabla f_i(x_i)$, and $w_{ij}$ is the $(i, j)$-th entry of the weight matrix $W$. Similar to ASBCD, node $i$ broadcasts $x_i$ to all neighbors, and its neighbor $j$ overwrites $x_{ji}$ in its buffer $\mathcal{B}_j$. A detailed implementation is given in Algorithm 2.

Likewise, the iterates are indexed by $k \in \mathbb{N}_0$, which increases by 1 whenever an update is performed on a local variable $x_i$ of some nodes $i \in \mathcal{V}$. Again, $k$ is only for analysis and need not be known by agents. Denote $\bar{\mathcal{N}}_i = \mathcal{N}_i \cup \{i\}$ for all $i \in \mathcal{V}$. Then, the asynchronous DSGD can be described as

$$x_i^{k+1} = \begin{cases} \sum_{j \in \bar{\mathcal{N}}_i} w_{ij} x_j^{s_{ij}^k} - \alpha g^{f_i}(x_i^k), & i = i_k, \\ x_i^k, & \text{otherwise,} \end{cases} \tag{6}$$

where $s_{ij}^k \in [0, k]$ for $j \in \bar{\mathcal{N}}_i$ is the largest iteration index (smaller than or equal to $k$) of the most recent version of $x_j$ available to node $i$ at iteration $k$ (instead of "when $i$ begins its final gradient estimation prior to iteration $k$" as in ASBCD). Still, $s_{ij}^k$ is the minimal value of the current iteration index $k$ and the time agent $j$ updates next as previously defined in ASBCD.

Similar schematics of $s_{ij}^k$ are shown in Fig. 1c and 1d. The behaviors of both agents in Fig. 1c are identical to that of Fig. 1a, and so is Fig. 1d to Fig. 1b. Now $s_{i_9 1}^9$ is the largest iteration index of available $x_1$ when agent $i_9$ finishes gradient estimation at iteration 9. Note that, in Fig. 1c, the $x_1$ updated at iteration 5 is available to agent $i_9$ prior to iteration 9, and $x_1$ remains identical until

---

**Algorithm 2** ADSGD

---

1: **Initialization:** All the nodes agree on $\alpha > 0$, and cooperatively set $w_{ij}, \forall \{i, j\} \in \mathcal{E}$.
2: Each node chooses $x_i$, creates a buffer $\mathcal{B}_i$, shares $x_i$ with neighbors, and calculates $g^{f_i}(x_i)$.
3: **All Nodes Do In Parallel:**
4: **while** termination criterion not met **do**
5:     **repeat**
6:         Keep receiving $x_j$ from neighbors.
7:         Let $x_{ij} = x_j$ and store $x_{ij}$ in $\mathcal{B}_i$
8:     **until** $g^{f_i}(x_i)$ is available.
9:     Update $x_i$ according to equation 5.
10:     Send $x_i$ to all neighbors $j \in \mathcal{N}_i$.
11:     Calculate $g^{f_i}(x_i)$.
12: **end while**

---

iteration 10. Therefore, $s_{i_9 1}^9 = \min\{10, 9\} = 9$ for Fig. 1c. Likewise, in Fig. 1d, the $x_1$ updated at iteration 3 is available to agent $i_9$ before iteration 9. Thus, $s_{i_9 1}^9 = \min\{8, 9\} = 8$ for Fig. 1d.

**Remark 2.2.** Unlike in ASBCD, $s_{ij}^k$ here captures only the communication delay. The difference originates from the time the global model is used. The global model is used before gradient estimation in ASBCD and after gradient estimation in ADSGD. When $i_k = [n]$ and $s_{ij}^k = k$, for all $i, j, k$, equation 6 reduces to the synchronous DSGD.

**Notation.** With a slight abuse of notation, $[x]_i$ refers to taking the $i$-th block of a vector $x$. For instance, if $\mathbf{x} \triangleq [x_1, ..., x_n]$, then $[\mathbf{x}]_i = x_i$, where $x_i$ can either be a scalar or a vector. For a matrix $W$ with $n$ eigenvalues, $\lambda_i(W)$ denotes the $i$-th largest eigenvalue. Additionally, we define $\mathbf{W} = W \otimes I_d$ and $(k)^+ \triangleq \max\{0, k\}$.

## 3 CONVERGENCE ANALYSIS

### 3.1 ASSUMPTIONS

All assumptions are summarized below. Assumption 3.1 - 3.2 are for ASBCD, while their counterparts Assumption 3.5 - 3.6 are for ADSGD. The assumption on partial asynchrony (Assumption 3.3) is shared.

**Assumption 3.1.** $\mathbf{f}$ *is $L$-smooth and lower bounded by* $\mathbf{f}^*$.

**Assumption 3.2.** *For each block $i$, the gradient estimator is unbiased with bounded variance; that is,* $\mathbb{E}[g_i^{\mathbf{f}}(\mathbf{x}) - \nabla_i \mathbf{f}(\mathbf{x})] = 0$ *and* $\mathbb{E}[\|g_i^{\mathbf{f}}(\mathbf{x}) - \nabla_i \mathbf{f}(\mathbf{x})\|^2] \leq \sigma^2, \forall i, \mathbf{x}$.

**Assumption 3.3** (Asynchrony)**.** *There exist positive integers $B$ and $D$ such that*

    *1. For every $i \in \mathcal{V}$ and for every $k \geq 0$, there exists $m \in \{k, \ldots, k+B-1\}$ such that $i_m = i$.*

    *2. There holds $k - D \leq s_{ij}^k \leq k$ for all $i \in \mathcal{V}$, $j \in \mathcal{N}_i$, and $k \in \mathcal{K}_i$.*

**Remark 3.4.** Assumption 3.3.1 requires each agent to update at least once every $B$ steps. Therefore, $B$ resembles the computation delay. Assumption 3.3.2 implies different conditions for ADSGD and ASBCD. For ADSGD, $D$ represents the maximum communication delay, as the agents mix the neighboring models after the gradient is available. Whereas for ASBCD, $D$ encompasses both computation and communication delay, as each block takes the whole model before gradient estimation. Therefore, when there is no communication delay, $D$ for ADSGD degenerates to 0 while $D$ for ASBCD degenerates to $B$. In practice, the assumption holds as long as the computation and communication times of each agent are lower and upper bounded.

**Assumption 3.5.** *For each agent, the objective function $f_i$ is $L_i$-smooth and lower bounded by $f_i^*$.*

**Assumption 3.6.** *For each agent, the gradient estimator is unbiased with bounded variance; that is,* $\mathbb{E}[g^{f_i}(x) - \nabla f_i(x)] = 0$ *and* $\mathbb{E}[\|g^{f_i}(x) - \nabla f_i(x)\|^2] \leq \sigma^2, \forall i, x$.

**Assumption 3.7.** *The mixing matrix $W$ is stochastic and symmetric, with the corresponding communication graph being undirected and connected.*

## 3.2 Convergence of ASBCD

The following lemma generalizes [Theorem 1, Sun et al. (2017)] to stochastic gradients, forming the basis for ADSGD's analysis:

**Lemma 3.8.** *For problem 2, given Assumption 3.1 - 3.3, and* $\alpha < \frac{1}{(D+1/2)L}$, *the sequence* $\{\mathbf{x}^k\}$ *generated by (4) satisfies the following relation:*

$$\frac{\sum_{k=0}^{K-1} \mathbb{E}\|\nabla \mathbf{f}(\mathbf{x}^k)\|^2}{K} \leq \frac{3n(B + C_0 L^2 \alpha^2)}{\alpha(1 - (\frac{L}{2} + DL)\alpha)} \frac{\mathbf{f}(\mathbf{x}^0) - \mathbf{f}^*}{K} + \alpha \left( 3nC_0 L^2 \alpha + \frac{L(D+1)}{2(1 - (\frac{L}{2} + DL)\alpha)} \right) \sigma^2,$$

*where* $C_0 = D^2 + 3B^2(D + 2D^3)$.

Lemma 3.8 indicates ASBCD converges to the $\mathcal{O}(\alpha)$ neighborhood of its stationary points with a rate of $\mathcal{O}(\frac{1}{K})$, matching the standard rate for non-convex SGD. Note that the step size here depends on $D$, encompassing computation and communication delays. A corollary is provided in the appendix.

## 3.3 Convergence of ADSGD

We now turn to the convergence analysis of ADSGD, which builds on the ASBCD framework but introduces additional challenges due to stochastic gradients and asynchrony.

Define $F(\mathbf{x}) \triangleq \sum_i f_i(x_i)$, and $L_\alpha(\mathbf{x}) = F(\mathbf{x}) + \frac{\mathbf{x}^T(I-\mathbf{W})\mathbf{x}}{2\alpha}$. Note that $F(\mathbf{x})$ and $L_\alpha(\mathbf{x})$ are both Lipschitz smooth with $L_F \triangleq \max L_i$ and $L_L \triangleq L_F + \frac{1-\lambda_n}{\alpha}$, respectively (note that $L_L$ is a function of the step size $\alpha$).

ADSGD on $F(\mathbf{x})$ can be viewed as ASBCD on $L_\alpha(\mathbf{x})$. The updating rule equation 6 can be rewritten as

$$x_i^{k+1} = \begin{cases} x_i^k - \alpha g_i^{L_\alpha}(\hat{\mathbf{x}}^k), & i = i_k, \\ x_i^k, & \text{otherwise}, \end{cases}$$

where $g_i^{L_\alpha}(\cdot)$ is a stochastic estimate of $\nabla_i L_\alpha(\cdot)$ and $\hat{\mathbf{x}}^k = ((x_1^{s_{i_k 1}^k})^T, ..., (x_n^{s_{i_k n}^k})^T)^T$. Note that even though the above is expressed under the framework of ASBCD, $s_{ij}^k$ should follow the one defined for ADSGD (Section 2.2). This is because, in ADSGD, the neighbors' models are only used after gradient estimation. Therefore, $\hat{\mathbf{x}}^k$ should be the global model available to $i_k$ at iteration $k$.

ADSGD can be viewed within the ASBCD framework, where the effective Lipschitz constant depends on the step size. This dependency complicates convergence analysis, since step size alone cannot control noise accumulation. To address this limitation, we propose a double-step-size technique, introducing an auxiliary step size $\beta$ such that

$$\begin{aligned} x_{i_k}^{k+1} &= x_{i_k}^k - \beta g_{i_k}^{L_\alpha}(\hat{\mathbf{x}}^k) \\ &= (1 - \frac{\beta}{\alpha})x_{i_k}^k + \frac{\beta}{\alpha}[\mathbf{W}\hat{\mathbf{x}}^k]_{i_k} - \beta g_{i_k}^F(\hat{\mathbf{x}}^k) \\ &= [\tilde{\mathbf{W}}\hat{\mathbf{x}}^k]_{i_k} - \beta g^{f_{i_k}}(x_{i_k}^k), \end{aligned} \tag{7}$$

where $\tilde{\mathbf{W}} \triangleq \tilde{W} \otimes I_d$, and $\tilde{W}_{ii} = (1 - \frac{\beta}{\alpha}) + \frac{\beta}{\alpha}W_{ii}$, $\tilde{W}_{ij} = \frac{\beta}{\alpha}W_{ij}, i \neq j$. The double-step size ADSGD algorithm is effectively Algorithm 2 with a different weight matrix $\tilde{W}$ and the step size $\beta$, since $\tilde{W}$ satisfies Assumption 3.7. Via the ASBCD-ADSGD correspondence, we transfer ASBCD's convergence to Algorithm 2. Theorem 3.9 shows convergence under proper $\{\alpha, \beta\}$.

**Theorem 3.9.** *For Problem (1), given Assumption 3.3 - 3.7 and* $\beta < \frac{1}{(D+1/2)L_L}$, *the sequence* $\{x^k\}$ *generated by equation 7 satisfies:*

$$\frac{\sum_{k=0}^{K-1} \mathbb{E}\|\nabla f(\bar{x}^k)\|^2}{K} \leq \frac{6n^2(B + C_0 L_L^2 \beta^2)}{\beta(1 - (D + \frac{1}{2})L_L \beta)} \frac{\sum_{i=1}^n (f_i(x^0) - f_i^*)}{K}$$

$$+ L_L \beta \left( 6n^2 C_0 L_L \beta + \frac{n(D+1)}{1 - (D + \frac{1}{2})L_L \beta} \right) \sigma^2$$

$$+ \frac{4n(\max_i L_i)^2 \alpha}{1 - \lambda_2(W)} \left( \sum_{i=1}^n (f_i(x^0) - f_i^*) + \frac{D+1}{2} K L_L \beta^2 \sigma^2 \right)$$

where $\bar{x}^k = \frac{1}{n} \sum_{i=1}^n x_i^k$ and $C_0$ is defined in Lemma 3.8.

**Remark 3.10.** Unlike (Zhu et al., 2023; Kungurtsev et al., 2023), the proposed step size $\beta$ is independent of computation delay $B$ and permits much larger values than prior asynchronous decentralized non-convex SGD methods. This independence arises because each agent i computes gradients solely using its local model $x_i^t$, which isn't affected by communication delays.

**Remark 3.11.** In contrast to previous convergence analyses of DSGD Koloskova et al. (2020), current analysis does not rely on bounded data heterogeneity assumption. It only requires each individual loss to be bounded below, which is almost always satisfied in practice.

**Corollary 3.12.** *For Problem (1), given Assumption 3.3 - 3.7 and let* $\alpha = \frac{2}{L_F K^{1/3}}, \beta = \frac{1}{4L_F(D+1/2)K^{2/3}},$ *the sequence* $\{x^k\}$ *generated by equation 7 satisfies:*

$$\frac{\sum_{k=0}^{K-1} \mathbb{E}\|\nabla f(\bar{x}^k)\|^2}{K} \leq \left(16n^2 C_1 + \frac{8n}{1 - \lambda_2(W)}\right) L_F \frac{\sum_{i=1}^n (f_i(x^0) - f_i^*)}{K^{1/3}}$$

$$+ \left(\frac{n}{D(1 - \lambda_2(W))} + 2n\right) \frac{\sigma^2}{K^{1/3}} + \frac{3n^2 C_0}{2D} \frac{\sigma^2}{K^{2/3}},$$

*where* $C_0$ *is defined in Lemma 3.8 and* $C_1 = 6B^2 D^2 + 3B^2 + 3BD + 3B + D$.

Corollary 3.12 establishes that ADSGD converges to stationary points with proper step sizes, albeit at a rate slightly slower than standard non-convex SGD due to consensus error ($\mathcal{O}(\alpha)$). While convergence to $L_\alpha$'s stationary points achieves the faster $\mathcal{O}(1/\sqrt{K})$ rate with a proper step size, the gap between $L_\alpha$ and $F$ limits the overall rate. The key challenge lies in controlling consensus error while maintaining full asynchrony. ADSGD's self-only update rule (Eq. 5) breaks double stochasticity, invalidating prior analyses. Partial neighbor synchronization Lian et al. (2018) preserves this property but introduces stalls and performance degradation (Section 4).

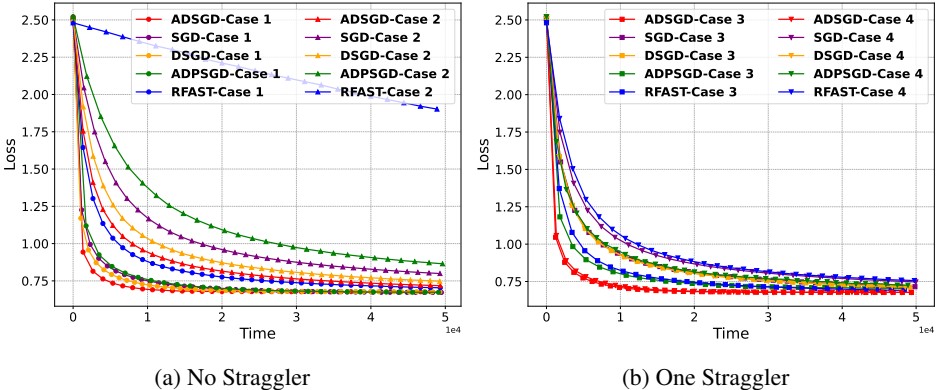

(a) No Straggler  (b) One Straggler

Figure 2: Loss plot of Logistic Regression on MNIST. Case 5 is deferred to Appendix B for clarity.

## 4 EMPIRICAL EVALUATION

To empirically demonstrate ADSGD's effectiveness, we compare it with ADPSGD (Lian et al., 2018) and RFAST (Zhu et al., 2023) (a gradient-tracking method), while using synchronous DSGD (Yuan et al., 2016) and parallel SGD (ring-Allreduce implementation) as baselines. For all algorithms, the implementation adheres strictly to their original descriptions, without incorporating any additional acceleration techniques. These algorithms are tested on two non-convex tasks with real-world datasets. Experiments on two non-convex tasks under diverse delay scenarios are conducted on a server with 8 Nvidia RTX3090 GPUs. All details are included in Appendix B.1.

**Modeling delays.** We simulate system delays via random sampling for better ablation (see Appendix B.1.2). Computation delays focus on gradient estimation (the dominant time cost), ignoring negligible model mixing/updating delays. For communication, we assume full-duplex agents with multicast and serial sending (i.e., sequential message transmission). These conservative assumptions hold in practice: full-duplex multicasting is supported by 4G, WiFi, Zigbee, and wired LANs. Serial sending improves information availability under bandwidth limits.

**Tasks description.** We conduct our main experiments using 9 agents connected in a grid network, and further evaluate scalability by extending the setup to as many as 128 agents. We first conduct logistic regression with non-convex regularization on the MNIST dataset LeCun et al. (1998). A modified VGG11 Simonyan & Zisserman (2014) is then trained over CIFAR-10 Krizhevsky et al. (2009). We evaluate the algorithms under varying degrees of data heterogeneity, but report only the most challenging case in the main text: the *fully label-partitioned* setting. This scenario is substantially more challenging than the commonly used Dirichlet (0.1) benchmark, which is already regarded as highly heterogeneous. Appendix B further shows that ADSGD's advantage widens as heterogeneity decreases. For each task & algorithm, there are 5 test cases as summarized in Table 1. Note communication and computation on average takes 1 unit of time in case 1) and communication is twice as slow in case 2) for VGG (slower communication delays convergence and increases computational overhead, thus not investigated). We report the main results that substantiate our claim, while a comprehensive presentation of all experiments is provided in Appendix B.

Table 1: Test Cases for All Algorithms

| Case | Description | Case | Description |
| --- | --- | --- | --- |
| 1) Base | Uniform delays | 2) Slow Comm. | $10\times(2\times)$ slower comm. |
| 3) Comp. Str. | 1 agent $10\times$ slower comp. | 4) Comm. Str. | 1 agent $10\times$ slower comm. |
| 5) Comb. Str. | 1 agent $10\times$ slower both | | |

**Parameter selection.** A fixed step size of 0.01 is adopted for both tasks and algorithms, except for RFAST, which uses $\frac{0.01}{w_{ii}}$ to match an identical effective step size. Local batch size for non-convex logistic regression and VGG training is set to 32 and 8, respectively.

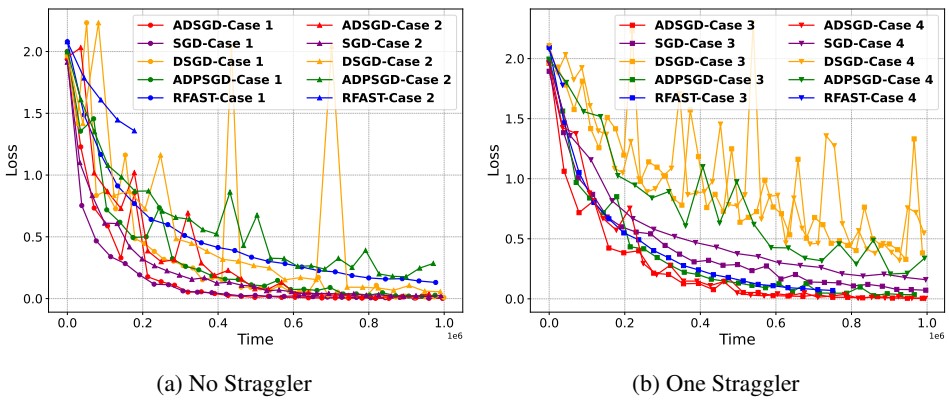

(a) No Straggler            (b) One Straggler

Figure 3: Loss plot of VGG on CIFAR10. Case 5 is deferred to Appendix B for clarity.

### 4.1 NON-CONVEX LOGISTIC REGRESSION ON MNIST

We present the loss curves of the average model $f(\bar{x})$ in Fig. 2. Without stragglers (Fig. 2a), ADSGD outperforms all other algorithms and uniquely surpasses the baselines. While asynchronous methods typically exhibit slower iteration-wise convergence due to stale information but faster per-iteration runtime, RFAST and ADPSGD fail to benefit from this trade-off. ADPSGD suffers from partial synchronization requirements that slow its runtime, whereas RFAST's neighbor-specific communications and doubled gradient tracking overhead make it particularly delay-sensitive.

Under the straggler scenarios (Fig. 2b), ADSGD maintains its lead. With communication stragglers, DSGD beats ADPSGD and RFAST, while ADSGD shows significantly stronger robustness. In computation delay cases (case 3), all asynchronous methods outperform the synchronous baseline, demonstrating their advantage under such conditions.

### 4.2 VGG11 ON CIFAR-10

This section demonstrates ADSGD's capabilities on more complex tasks. Following the previous methodology, Fig. 3a presents the average model's loss curve. All algorithms show convergence

patterns similar to logistic regression. Parallel SGD is the fastest in cases without straggler, but as a centralized baseline it avoids the challenges of data heterogeneity. Among the remaining methods, ADSGD maintains the fastest convergence. Notably, RFAST diverges in the case of Slow Comm., where the communication delay is only twice the computation delay. Though unstable, RFAST does exhibit much smaller loss oscillations due to gradient tracking.

**Remark 4.1.** Gradient tracking methods achieve $\mathcal{O}(1/\sqrt{K})$ convergence, matching the stochastic optimal rate. However, this comes at the cost of (i) requiring extremely small step sizes and (ii) inferior practical performance, as exemplified by RFAST: high delay-sensitivity due to doubled communication and substantial memory overhead (see Appendix A).

Fig. 3b confirms ADSGD's superiority. Asynchronous methods generally handle computation delays well, with ADSGD particularly robust against communication delays. RFAST diverges in all cases due to its delay sensitivity under non-convexity. Quantitatively, ADSGD achieves 85% test accuracy 15-70% faster than async methods (70% for comm. stragglers), 30-85% faster than sync methods, and 35-58% faster than parallel SGD under stragglers (See Appendix B).

**Remark 4.2.** Considering both synchronous and asynchronous methods, ADSGD is especially advantageous in the presence of stragglers. This property is particularly relevant for heterogeneous compute clusters, where it is increasingly common to mix different generations of GPUs (e.g., xAI's Colossus supercomputer, which combines NVIDIA H100s and GB200s).

## 4.3 SCALABILITY

While prior results demonstrate ADSGD's effectiveness, computational constraints previously limited the number of agents tested. ADSGD's scalability is presented by measuring speedup across increasing agent counts under a ring topology and case 1 (Fig. 4). Logistic Regression (Fig. 4a) scales to 128 agents, with loss curves showing consistent acceleration as parallelism grows. VGG11 Training (Fig. 4b) achieves comparable speedup up to 32 agents, beyond which hardware limitations prevent further testing. The near-linear trend in both tasks confirms ADSGD's ability to leverage distributed training efficiently, revealing its potential to large scale applications.

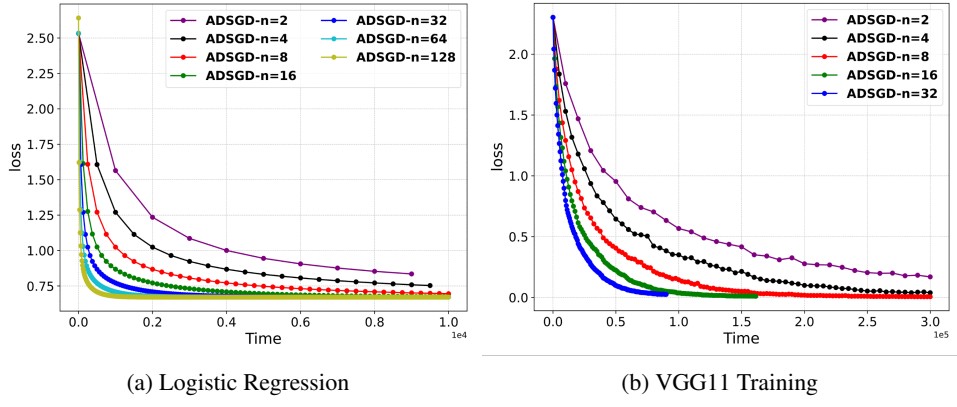

(a) Logistic Regression         (b) VGG11 Training

Figure 4: Speedup of ADSGD w.r.t. number of agents under a ring topology.

## 5 CONCLUSION

This paper explores the convergence of asynchronous algorithms under bounded communication and computation delays, focusing on ASBCD and ADSGD. We show that ASBCD converges to the neighborhood of its stationary points in non-convex settings and achieves a rate of $\mathcal{O}(1/\sqrt{K})$ given a proper step size. Extending these results to ADSGD, we prove its convergence under non-convexity with a computation-delay-independent step size, without assuming data heterogeneity. The experimental results confirm the effectiveness of ADSGD on non-convex learning tasks. We highlight that the proposed approach is simple, memory-efficient, communication-efficient, and highly resilient to communication delays, providing greater flexibility and robustness in decentralized optimization scenarios.

## REPRODUCIBILITY STATEMENT

We have taken deliberate steps to ensure the reproducibility of our results. All theoretical claims are accompanied by complete proofs in Appendix C. For empirical results, we provide detailed descriptions of datasets (MNIST, CIFAR-10), model architectures (logistic regression, VGG11), hyperparameters (learning rates, batch sizes, step sizes), and training protocols. We also specify the compute resources used, including GPU types and cluster configurations, as well as the simulation framework for modeling communication delays (see Appendix B.1). These materials will allow researchers to reproduce both the convergence plots and the scalability results reported in the paper.

## ETHICS STATEMENT

This work focuses on the development and analysis of decentralized optimization algorithms. Our contributions are methodological and theoretical in nature, and we do not foresee direct ethical risks from the algorithms themselves. The datasets used (MNIST and CIFAR-10) are standard, publicly available benchmarks that do not contain sensitive or personally identifiable information. While decentralized learning can enhance privacy by avoiding central data aggregation, it may also be deployed in settings where fairness, robustness, or misuse could become concerns. We encourage practitioners to carefully evaluate these aspects when applying our methods in real-world systems. Beyond these considerations, we are not aware of any immediate ethical issues raised by this research.

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

# A    ALGORITHM COMPARISON

Table 2 represents all decentralized stochastic gradient algorithms with provable convergence results under identical asynchrony assumptions. We present details of derivation in the following subsections.

Table 2: Decentralized stochastic gradient algorithms that converge under identical asynchrony assumptions. Memory and communication budget are normalized w.r.t. the model size. Comments: (1) SC and NC represent strong convexity and general non-convexity, respectively. (2) The step size in (Spiridonoff et al., 2020) is computation-delay-independent. However, they used diminishing step size, which will eventually be small enough for their convergence results. Readers may refer to the appendix for details.

| | Convexity[1] | Comp.-delay-ind. Step Size | Memory Cost | Comm. Budget |
|---|---|---|---|---|
| Spiridonoff et al. (2020) | SC | ✓[2] | $3|\mathcal{N}_i| + 3$ | $|\mathcal{N}_i|$ |
| Zhu et al. (2023) | NC | | $4|\mathcal{N}_i| + 5$ | $2|\mathcal{N}_i|$ |
| Kungurtsev et al. (2023) | NC | | $4|\mathcal{N}_i| + 5$ | $2|\mathcal{N}_i|$ |
| Ours | NC | ✓ | $|\mathcal{N}_i| + 2$ | $|\mathcal{N}_i|$ |

## A.1    ON STEP SIZES

Only (Spiridonoff et al., 2020) and this work adopt a computation-delay-independent step size. However, in (Spiridonoff et al., 2020), they use a diminishing step size rule to reach an asymptotic result, whereas we use a fixed step size.

As mentioned in Section 1.1, current gradient tracking methods require not only computation-delay-dependent step sizes but also extremely small ones. We simplify (increase) their required step sizes to illustrate how impractical their bounds are.

From Theorem III.1 in (Kungurtsev et al., 2023), its initial step size is upper bounded by $\frac{1}{4(L_F + \tilde{C}_1 + \tilde{C}_2)}$, where $\tilde{C}_1 \triangleq \frac{2\tilde{C}_0}{1-\rho^2}$, $\tilde{C}_0 \triangleq \frac{2\sqrt{(D+2)n}(1+\underline{w}^{-K_A})}{1-\underline{w}^{K_A}}$, $\rho \triangleq (1 - \underline{w}^{K_A})^{1/K_A}$, and $K_A \triangleq (2n-1)B + nD$. For ease of illustration, we increase the upper bound to $\frac{1}{\tilde{C}_1} = \frac{(1-\rho)^2}{2\tilde{C}_0}$. Note that this is a very loose bound since $\tilde{C}_2$ ($C_2^t$ in (Kungurtsev et al., 2023)) can be magnitudes larger than $\tilde{C}_1$.

From (39) in (Zhu et al., 2023), the step size is upper bounded by $\frac{r\eta^2}{16(1+6\rho_t)K}$, where $r$ is the number of common roots between subgraphs for pushing and pulling. For the ease of illustration, we assume that $r \leq 96$. Thus, the upper bound is now $\frac{\eta^2}{\rho_t K}$, where $\eta \triangleq \underline{w}^{K_A}$, $\rho_t \triangleq \frac{54\tilde{C}_3^2 C_L^2[4\tilde{C}_0^2+(1-\rho)^2]}{(1-\rho)^4}$, $\tilde{C}_3 \triangleq \frac{2\sqrt{2S}(1+\underline{w}^{-K_A})}{\rho(1-\underline{w}^{K_A})}$, $C_L = \max\{L_i\}$, and $S \geq (D+1)n$. Again for illustration, we assume $216\tilde{C}_3^2 C_L^2 \geq 1$, which should be satisfied for most applications. Therefore, we have $\rho_t \geq \frac{C_2^2}{(1-\rho)^4}$ and the step size must be smaller than $\frac{\eta^2(1-\rho^4)}{\tilde{C}_0^2 K}$.

## A.2    ON MEMORY AND COMMUNICATION COSTS

For (Spiridonoff et al., 2020), from their Algorithm 3, each agent stores $\{\mathbf{x}_i, \hat{\mathbf{g}}_i, \boldsymbol{\phi}_i^x\}$ for itself, and $\{\boldsymbol{\phi}_i^x, \boldsymbol{\rho}_{ij}^{*x}, \boldsymbol{\rho}_{ij}^x\}$ for all neighbors. In each iteration, each agent sends $\{\boldsymbol{\phi}_i^x\}$. Some scalars are ignored.

For (Zhu et al., 2023), each agent stores $\{x_i^t, z_i^t, v_i^t, \nabla f_i(x_i^{t+1}; \zeta_i^{t+1}), \nabla f_i(x_i^t; \zeta_i^t)\}$ for itself, and $\{v_j^{\tau_{v,ij}^t}, \rho_{ij}^{\tau_{\rho,ij}^t}, \tilde{\rho}_{ij}^t, \rho_{ji}^t\}$ for all neighbors. In each iteration, each agent sends $\{v_i^{t+1}, \rho_{ji}^{t+1}\}$. The calculation for (Kungurtsev et al., 2023) is identical.

In the proposed method, each agent only stores $\{x_i, g^{f_i}(x_i)\}$ for itself and $\{x_j\}$ for its neighbors. In each iteration, each agent sends $\{x_i\}$.

---

**Algorithm 3** Memory-efficient ADSGD

---

1: **Initialization:** All the nodes agree on $\alpha > 0$, and cooperatively set $w_{ij} \ \forall \{i, j\} \in \mathcal{E}$.
2: Each node chooses $x_i \in \mathbb{R}^d$, creates a local buffer $\mathcal{B}_i$, shares $x_i$ with all neighbors in $\mathcal{N}_i$, and calculates $g^{f_i}(x_i)$.
3: Each node stores $y_i = \sum_{j \in \mathcal{N}_i} w_{ij} x_j$ in $\mathcal{B}_i$.

4: **All Nodes Do In Parallel:**
5: **while** termination criterion not met **do**
6:     **repeat**
7:         Keep receiving $z_j$ from neighbors.
8:         Update $\mathcal{B}_i$ by $y_i = y_i + w_{ij} z_j$.
9:     **until** $g^{f_i}(x_i)$ is available.
10:     Send $z_i = (w_{ii} - 1)x_i + y_i - \alpha g^{f_i}(x_i)$ to all neighbors $j \in \mathcal{N}_i$.
11:     $x_i = x_i + z_i$.
12:     Calculates $g^{f_i}(x_i)$.
13: **end while**

---

### A.3 MEMORY-EFFICIENT IMPLEMENTATION OF ADSGD

In Table 2, ADSGD requires $|\mathcal{N}|_i + 2$ units of memory, scaling linearly with the number of neighbors. We can remove such a dependency by a straightforward memory-efficient implementation, as shown in Algorithm 3. The key idea is to store only the weighted sum of neighboring models, rather than each individual model. During each iteration, nodes exchange model updates instead of the full models. Consequently, each node only maintains $\{x_i, y_i, z_i, g^{f_i}(x_i)\}$, where $z_i$ is the model update. Note that since we introduced the additional variable $z_i$, the memory requirement becomes fixed at four times the model size, regardless of the number of neighbors $|\mathcal{N}|_i$. Such an implementation is beneficial when $|\mathcal{N}|_i > 2$.

## B  MORE ON EXPERIMENTS

### B.1  EXPERIMENT SETTINGS

#### B.1.1  COMPUTE RESOURCES

The experiments are conducted on a server with 8 NVIDIA 3090 GPUs, each with 24GB memory. The overall experiment takes roughly over 1200 GPU hours. The heavy computation can be attributed to two reasons: (1) We investigate 5 delay configurations and 3 levels of data heterogeneity. (2) RFAST is over twice slower than ADSGD due to its complicated update mechanism.

#### B.1.2  SIMULATION FRAMEWORK

Our experiments employ a discrete-event simulation framework designed to accurately model asynchronous distributed environments. Each agent operates independently, with its own computation and storage (local buffer), and no shared memory or computation across agents. Asynchrony is modeled via a priority queue: when an agent begins a task at time $t$ (e.g., gradient computation), it samples a delay $d$ from a distribution, and the corresponding event (e.g., gradient_ready) is scheduled at time $t + d$. The simulator then processes events in chronological order. This setup is both realistic and flexible, producing results equivalent to distributing tasks across multiple independent machines while enabling precise and reproducible ablation studies. For example, stragglers can be modeled simply by altering their delay distribution, offering more controlled analysis than prior approaches (e.g., RFAST, which simulates stragglers by adding extra computation). As mentioned in Section 4, computation and communication delays are simulated using random draws from distributions fitted to real-world GPU and interconnect measurements. We control only the mean of each distribution, while other parameters vary accordingly; in practice, communication delays exhibit larger variance than computation delays.

### B.1.3 TEST CASES

We elaborate on the 5 test cases mentioned in Table 1. 1) the base case with identical communication and computation speeds across all agents, where the mean of each delay distribution is set to 1; 2) the slow communication case, where all agents' communication is 10 times slower (2 times for the case of VGG training) than their computation; 3) the computation straggler case with one agent computes 10 times slower than others, while the remaining agents have identical computation and communication speed; 4) the communication straggler case with one agent communicates 10 times slower than others; and 5) a combined straggler case where one agent is 10 times slower in both communication and computation. Here, having identical communication and computation speeds indicates that the delay distributions have identical means while being different in shape. 10x slower computation implies the mean of computation delay is 10 times larger than the mean of communication delay, and vice versa.

### B.1.4 DATA HETEROGENEITY

We evaluate ADSGD under varying levels of data heterogeneity, quantified by parameter $\zeta$. Specifically, $\zeta$ denotes the fraction of each local dataset that is partitioned in a label-skewed manner, with the remainder sampled uniformly. Larger $\zeta$ values correspond to higher heterogeneity. For example, when $\zeta = 0$, data are partitioned uniformly across clients (homogeneous/IID). When $\zeta = 0.5$, each client receives a mix of label-skewed and uniformly sampled data, resulting in partial overlap but noticeable skew. When $\zeta = 1$, data are fully label-partitioned, i.e., each client holds only a subset of labels, yielding the most heterogeneous and challenging setting, which is presented in the main text.

Table 3 reports the relative time required to reach 83% test accuracy when training VGG11 on CIFAR-10 under different levels of data heterogeneity. As expected, training becomes progressively slower as heterogeneity increases: compared to the homogeneous case ($\zeta = 0$), the commonly used Dirichlet($\alpha = 0.1$) benchmark already incurs a 48% slowdown, while the fully label-partitioned setting ($\zeta = 1$) is more than 3.5 times slower. This highlights the severity of the fully partitioned regime adopted in the main text, which is substantially more challenging than standard benchmarks.

Table 3: Relative time to reach 83% test accuracy when training VGG11 on CIFAR-10 under different data heterogeneity settings (lower is better).

| Data Heterogeneity Setting | Relative Time to 83% Acc. |
| --- | --- |
| Homogeneous ($\zeta = 0$) | 1.00 |
| High (Dirichlet, $\alpha_{\text{Dir}} = 0.1$) | 1.48 |
| Extremely Heterogeneous ($\zeta = 1$) | 3.70 |

### B.2 NON-CONVEX LOGISTIC REGRESSION ON MNIST

Fig. 5 and 6 present the plots of training loss as a function of runtime for different algorithms. We see that under the presence of a combined straggler (Fig. 6c), parallel SGD performs the worst and the lead of ADSGD is considerable.

Fig. 7 and 8 present the plots of test accuracy as a function of runtime for different algorithms. Under all cases, ADSGD consistently achieves higher test accuracy within the same time frame. Notably, in most cases, asynchronous algorithms other than ADSGD even fail to outperform DSGD, rendering them of limited practical value.

Fig. 9 illustrates the relative time required to reach 89% test accuracy across different algorithms. ADSGD maintains a substantial lead, converging faster than other asynchronous algorithms by at least 50% in all cases. In case 4, ADSGD demonstrates its resilience to communication delay, saving over 85% of the time compared to RFAST, which is more than 7 times faster. In the case of slow communication, RFAST, while not diverging, converges very slowly and fails to reach 89% accuracy within $2 \times 10^5$ units of time (for comparison, ADSGD achieves this at $5.3 \times 10^4$ units of time). Moreover, ADSGD outpaces its synchronous counterpart by a margin of 28% - 75%.

It is important to note that ADPSGD cannot achieve 89% test accuracy in the presence of a straggler. This is because it converges to the stationary point of the weighted sum of local cost functions,

$\sum_i p_i f_i$, instead of the global objective $\sum_i f_i$, where $p_i$ represents the update frequency. This issue stems from its partially synchronized update $X_{k+1} = X_k W_k - \alpha g(\hat{\mathbf{x}}_k)$. For instance, in an extreme case where one agent is connected to all others and continues updating, that agent will dominate the updates and drag the other agents toward its own stationary point. Therefore, the partial synchronization in ADPSGD not only slows down the protocol under communication delays because of stalls but also introduces some bias into the optimization process.

Fig. 10 shows convergence of Logistic Regression under varying heterogeneity levels ($\zeta \in 0, 0.5$). ADSGD demonstrates stronger performance with lower $\zeta$ values, consistently outperforming baselines across all scenarios.

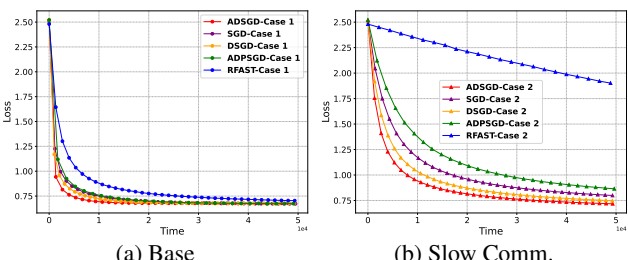

(a) Base                              (b) Slow Comm.

Figure 5: Logistic Regression - No Straggler - Training Loss - $\zeta = 1$

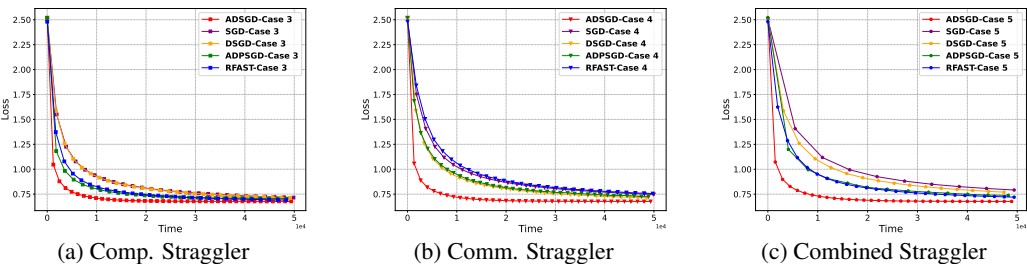

(a) Comp. Straggler          (b) Comm. Straggler          (c) Combined Straggler

Figure 6: Logistic Regression - One Straggler - Training Loss - $\zeta = 1$

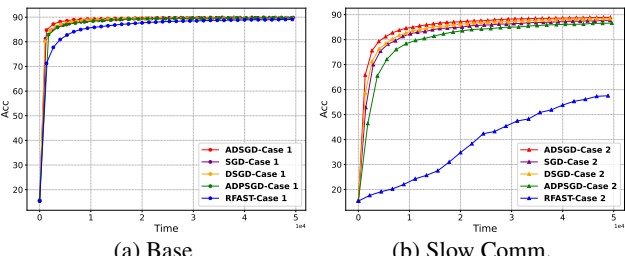

(a) Base                              (b) Slow Comm.

Figure 7: Logistic Regression - No Straggler - Test Accuracy - $\zeta = 1$

## B.3 VGG11 ON CIFAR-10

Fig. 11 and 12 present the training loss w.r.t. runtime for different algorithms. With a combined straggler (Fig. 12c), ADSGD maintains its lead by a significant amount, consistent with its performance in logistic regression.

Fig. 13 and 14 present the test accuracy w.r.t. runtime for different algorithms. When there is no straggler, parallel SGD reaches a higher test accuracy given the same amount of time, followed by ADSGD. Note that the loss function of VGG model is highly non-convex, resulting in additional difficulties on adopting stale information and dealing with data heterogeneity. Thus, parallel SGD beats all algorithms under certain cases. When there is a straggler, ADSGD consistently outperforms all other algorithms in most scenarios, except in the case of computation stragglers, where asynchronous algorithms show comparable performance. The accuracy drops observed in RFAST under several conditions highlight its sensitivity to delays.

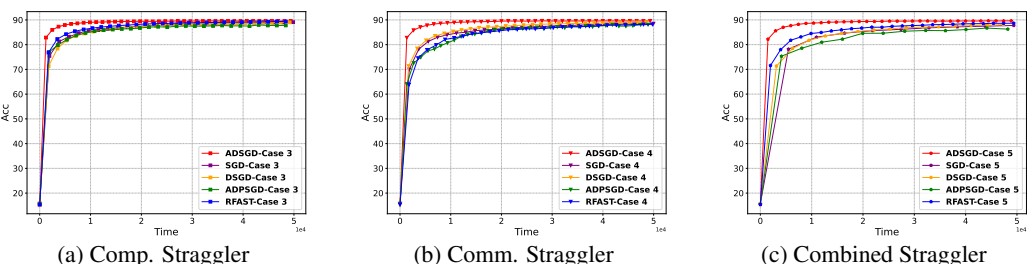

(a) Comp. Straggler  (b) Comm. Straggler  (c) Combined Straggler

Figure 8: Logistic Regression - One Straggler - Test Accuracy - $\zeta = 1$

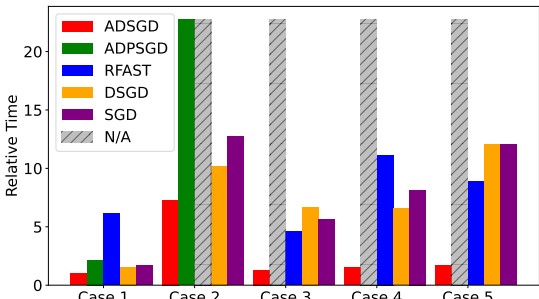

Figure 9: Relative time (**lower is better**) to achieve 89% test accuracy for Non-convex Logistic Regression on MNIST, normalized w.r.t. the runtime of ADSGD Case 1). N/A indicates the algorithm did not reach 89% accuracy. $\zeta = 1$

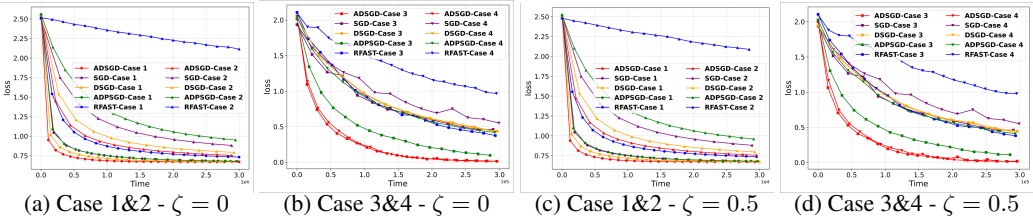

(a) Case 1&2 - $\zeta = 0$  (b) Case 3&4 - $\zeta = 0$  (c) Case 1&2 - $\zeta = 0.5$  (d) Case 3&4 - $\zeta = 0.5$

Figure 10: Logistic Regression under smaller data heterogeneity levels

We quantify the advantage of ADSGD using the relative runtime to achieve 85% test accuracy. As shown in Fig. 15, ADSGD saves at least 15% of the time compared to other asynchronous algorithms. In the case of comm. straggler, ADSGD saves over 70% of the time compared to other asynchronous methods. ADSGD also saves from 30% to 85% of the time against its synchronous counterpart. Additionally, ADSGD outpaces parallel SGD by 35% to 58% under the straggler condition.

Fig. 16 shows convergence for VGG training under different heterogeneity levels ($\zeta \in 0, 0.5$). Mirroring the logistic regression results, ADSGD achieves better performance with lower $\zeta$ values, consistently surpassing all baselines.

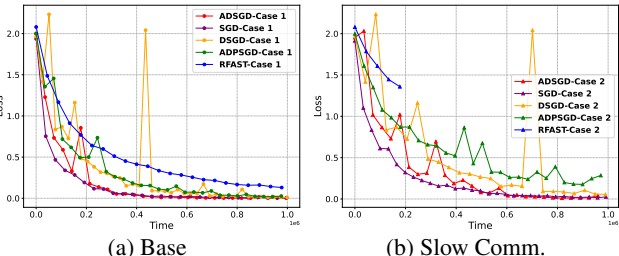

(a) Base      (b) Slow Comm.

Figure 11: VGG - No Straggler - Training Loss - $\zeta = 1$

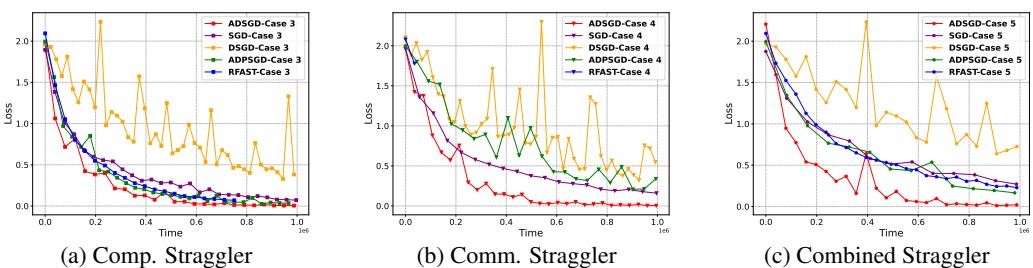

(a) Comp. Straggler      (b) Comm. Straggler      (c) Combined Straggler

Figure 12: VGG - One Straggler - Training Loss - $\zeta = 1$

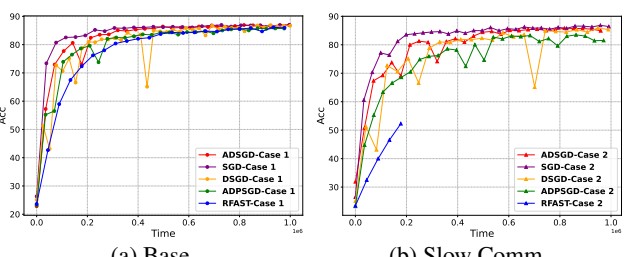

(a) Base      (b) Slow Comm.

Figure 13: VGG - No Straggler - Test Accuracy - $\zeta = 1$

## C   CONVERGENCE PROOF

### C.1   PROOF OF LEMMA 3.8

Following a similar way as in (Sun et al., 2017), let

$$\xi_f^k \triangleq \mathbf{f}(\mathbf{x}^k) + \frac{L}{2\epsilon} \sum_{i=(k-D)^+}^{k-1} (i - (k - D) + 1)\|\Delta^i\|^2,$$

where $\Delta^k \triangleq \mathbf{x}^{k+1} - \mathbf{x}^k = -\alpha g_{i_k}^{\mathbf{f}}(\hat{\mathbf{x}}^k)$, and we define $d^k \triangleq \mathbf{x}^k - \hat{\mathbf{x}}^k$.

We first characterize the relation between $\Delta^k$ and $d^k$. The following proof follows exactly the reasoning in (Zhou et al., 2018). However, we found that such a relation holds for a broader class of algorithms.

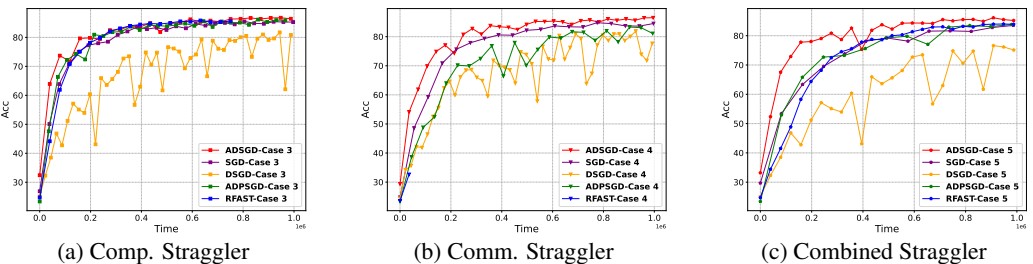

(a) Comp. Straggler     (b) Comm. Straggler     (c) Combined Straggler

Figure 14: VGG - One Straggler - Test Accuracy - $\zeta = 1$

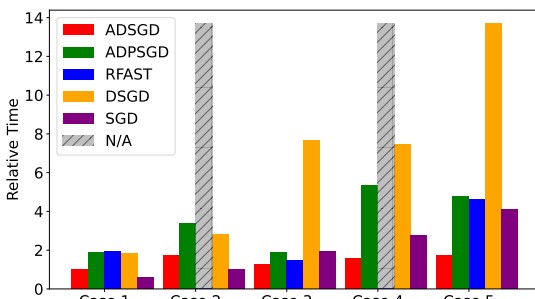

Figure 15: Relative time (**lower is better**) to achieve 85% test accuracy for VGG11 on CIFAR-10, normalized w.r.t. the runtime of ADSGD Case 1). N/A indicates the algorithm did not reach 85% accuracy. $\zeta = 1$

**Lemma C.1.** *Let $x_i^k \in \mathbb{R}^{d_i}$ and $\mathbf{x}^k = (x_1^k, ..., x_n^k) \in \mathbb{R}^{d'}$. For any algorithm that updates in the following form*

$$x_i^{k+1} = \begin{cases} T_i(\hat{\mathbf{x}}^k), & i = i_k, \\ x_i^k, & otherwise, \end{cases}$$

*where $\hat{\mathbf{x}}^k = (x_1^{t_1^k}, ..., x_n^{t_n^k})$ and $T_i(\cdot)$ is some mapping from $\mathbb{R}^{d'}$ to $\mathbb{R}^{d_i}$, if $k - t_i^k \leq D$ for all $i$ and $k$, we have*

$$\|\mathbf{x}^k - \hat{\mathbf{x}}^k\| \leq \sum_{t=(k-D)+}^{k-1} \|\mathbf{x}^{t+1} - \mathbf{x}^t\|.$$

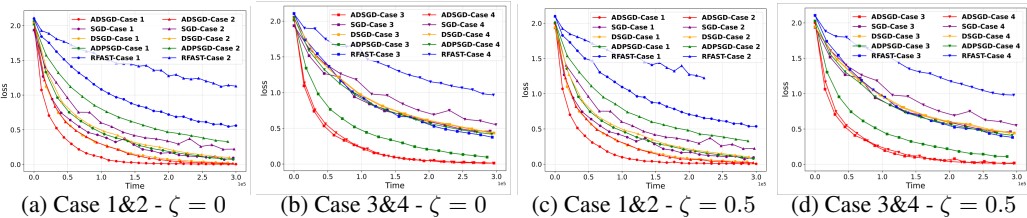

(a) Case 1&2 - $\zeta = 0$    (b) Case 3&4 - $\zeta = 0$    (c) Case 1&2 - $\zeta = 0.5$    (d) Case 3&4 - $\zeta = 0.5$

Figure 16: VGG training under smaller data heterogeneity levels

*Proof.*

$$\|\mathbf{x}^k - \hat{\mathbf{x}}^k\|^2 = \sum_{i=1}^n \|x_i^k - x_i^{t_i^k}\|^2$$

$$\leq \sum_{i=1}^n \left( \sum_{t=t_i^k}^{k-1} \|x_i^{t+1} - x_i^t\| \right)^2$$

$$\leq \sum_{i=1}^n \left( \sum_{t=(k-D)^+}^{k-1} \|x_i^{t+1} - x_i^t\| \right)^2$$

$$= \sum_{i=1}^n \sum_{t=(k-D)^+}^{k-1} \sum_{t'=(k-D)^+}^{k-1} \|x_i^{t+1} - x_i^t\| \|x_i^{t'+1} - x_i^{t'}\|$$

$$= \sum_{t=(k-D)^+}^{k-1} \sum_{t'=(k-D)^+}^{k-1} \sum_{i=1}^n \|x_i^{t+1} - x_i^t\| \|x_i^{t'+1} - x_i^{t'}\|$$

$$\leq \sum_{t=(k-D)^+}^{k-1} \sum_{t'=(k-D)^+}^{k-1} \|\mathbf{x}^{t+1} - \mathbf{x}^t\| \|\mathbf{x}^{t'+1} - \mathbf{x}^{t'}\|$$

$$= \left( \sum_{t=(k-D)^+}^{k-1} \|\mathbf{x}^{t+1} - \mathbf{x}^t\| \right)^2$$

$\square$

**Remark C.2.** Lemma C.1 bounds the staleness error by the sum of magnitude of updates. It holds for any algorithm that obeys a block-updating pattern with staled parameters, e.g. ASBCD and ADSGD.

By Lemma C.1, we have

$$\|d^k\| \leq \sum_{t=(k-D)^+}^{k-1} \|\Delta^k\| \tag{8}$$

With the above relation, we are ready to prove the following.

**Lemma C.3.** *Given Assumption 3.1 - 3.3, and $\alpha < \frac{1}{(D+\frac{1}{2})L}$, the sequence $\{\mathbf{x}^k\}$ generated by (4) satisfies the following relation:*

$$\frac{\sum_{k=0}^{K-1} \mathbb{E}\|\nabla_{i_k}\mathbf{f}(\hat{\mathbf{x}}^k)\|^2}{K} \leq \frac{\mathbf{f}(\mathbf{x}^0) - \mathbf{f}^*}{\alpha(1 - (D+\frac{1}{2})L\alpha)K} + \frac{(D+1)L\alpha}{2(1 - (D+\frac{1}{2})L\alpha)}\sigma^2$$

*Proof.* We have

$$\langle \mathbb{E}\Delta^k, \nabla f(\hat{\mathbf{x}}^k) \rangle = -\alpha\|\nabla_{i_k}f(\hat{\mathbf{x}}^k)\|^2 = -\frac{1}{\alpha}\|\mathbb{E}\Delta^k\|^2, \tag{9}$$

and

$$Var(\Delta^k) = \alpha^2\sigma^2, \tag{10}$$

where the expectation $\mathbb{E}$ is taken over the randomness of the gradient estimator.

By $L$-smoothness,

$$\mathbf{f}(\mathbf{x}^{k+1}) \leq \mathbf{f}(\mathbf{x}^k) + \langle \nabla\mathbf{f}(\mathbf{x}^k), \Delta^k \rangle + \frac{L}{2}\|\Delta^k\|_2^2. \tag{11}$$

We define the filtration $\mathcal{F}_k$ as a sequence of $\sigma$-algebra that captures all the randomness up to and including the $k$-th iteration. Take conditional expectation of equation 11, by Assumption 3.2,

$$\mathbb{E}[\mathbf{f}(\mathbf{x}^{k+1})|\mathcal{F}_k] - \mathbf{f}(\mathbf{x}^k) \overset{equation\ 9}{\leq} \langle \nabla\mathbf{f}(\mathbf{x}^k) - \nabla\mathbf{f}(\hat{\mathbf{x}}^k), \mathbb{E}[\Delta^k|\mathcal{F}_k]\rangle + \frac{L}{2}\mathbb{E}[\|\Delta^k\|^2|\mathcal{F}_k] - \frac{1}{\alpha}\|\mathbb{E}[\Delta^k|\mathcal{F}_k]\|^2$$

$$\leq L\|d^k\| \cdot \|\mathbb{E}[\Delta^k|\mathcal{F}_k]\| + \frac{L}{2}\mathbb{E}[\|\Delta^k\|^2|\mathcal{F}_k] - \frac{1}{\alpha}\|\mathbb{E}[\Delta^k|\mathcal{F}_k]\|^2$$

$$\leq \frac{L}{2\epsilon}\sum_{i=(k-D)^+}^{k-1}\|\Delta^i\|^2 + (\frac{\epsilon LD}{2} - \frac{1}{\alpha})\|E[\Delta^k|\mathcal{F}_k]\|^2 + \frac{L}{2}\mathbb{E}[\|\Delta^k\|^2|\mathcal{F}_k],$$

where the last inequality is from equation 8.

Therefore, by definition of $\xi_f^k$,

$$\xi_f^k - \mathbb{E}[\xi_f^{k+1}|\mathcal{F}_k] \geq \mathbf{f}(\mathbf{x}^k) - \mathbb{E}[\mathbf{f}(\mathbf{x}^{k+1})|\mathcal{F}_k] + \frac{L}{2\epsilon}\sum_{i=(k-D)^+}^{k-1}\|\Delta^i\|^2 - \frac{LD}{2\epsilon}\mathbb{E}[\|\Delta_k\|^2|\mathcal{F}_k]$$

$$\geq (\frac{1}{\alpha} - \frac{\epsilon LD}{2})\|\mathbb{E}[\Delta^k|\mathcal{F}_k]\|^2 - (\frac{LD}{2\epsilon} + \frac{L}{2})\mathbb{E}[\|\Delta^k\|^2|\mathcal{F}_k]$$

$$= (\frac{1}{\alpha} - \frac{\epsilon LD}{2} - \frac{LD}{2\epsilon} - \frac{L}{2})\|\mathbb{E}[\Delta^k|\mathcal{F}_k]\|^2 - (\frac{LD}{2\epsilon} + \frac{L}{2})Var(\Delta^k)$$

$$\overset{\epsilon=1}{\geq} (\frac{1}{\alpha} - \frac{L}{2} - DL)\alpha^2\|\nabla_{i_k}\mathbf{f}(\hat{\mathbf{x}}^k)\|^2 - \frac{L(D+1)}{2}\alpha^2\sigma^2 \qquad (12)$$

Take full expectation of the above and sum over $k$,

$$(\frac{1}{\alpha} - \frac{L}{2} - DL)\alpha^2 \frac{\sum_{k=0}^{K-1}\mathbb{E}\|\nabla_{i_k}\mathbf{f}(\hat{\mathbf{x}}^k)\|^2}{K} \leq \frac{\xi_f^0 - \xi_f^K}{K} + \frac{L(D+1)}{2}\alpha^2\sigma^2$$

$$\leq \frac{\mathbf{f}(\mathbf{x}^0) - \mathbf{f}^*}{K} + \frac{L(D+1)}{2}\alpha^2\sigma^2$$

$\square$

Lemma C.3 characterizes the convergence of $\mathbb{E}\|\nabla_{i_k}f(\hat{\mathbf{x}}^k)\|^2$, which implies the convergence of the ABCD algorithm. Based on the lemma, we now derive the convergence rate of $\mathbb{E}\|\nabla f(\mathbf{x}^k)\|^2$.

Note that

$$E\|\nabla_i\mathbf{f}(\mathbf{x}^k)\|^2 \leq 3(\mathbb{E}\|\nabla_i\mathbf{f}(\hat{\mathbf{x}}^{t_i(k)})\|^2 + \mathbb{E}\|\nabla_i\mathbf{f}(\mathbf{x}^k) - \nabla_i\mathbf{f}(\hat{\mathbf{x}}^k)\|^2 + \mathbb{E}\|\nabla_i\mathbf{f}(\hat{\mathbf{x}}^k) - \nabla_i\mathbf{f}(\hat{\mathbf{x}}^{t_i(k)})\|^2)$$

$$\leq 3(\mathbb{E}\|\nabla_i\mathbf{f}(\hat{\mathbf{x}}^{t_i(k)})\|^2 + L^2\mathbb{E}\|d^k\|^2 + L^2B\sum_{j=t_i(k)}^{k-1}\mathbb{E}\|\hat{\mathbf{x}}^{j+1} - \hat{\mathbf{x}}^j\|^2). \qquad (13)$$

For the second term on the RHS,

$$\mathbb{E}\|d^k\|^2 \leq D\sum_{j=(k-D)^+}^{k-1}\mathbb{E}\|\Delta^j\|^2$$

$$= D\alpha^2\sum_{j=(k-D)^+}^{k-1}(\sigma^2 + \mathbb{E}\|\nabla_{i_j}\mathbf{f}(\hat{\mathbf{x}}^j)\|^2). \qquad (14)$$

For the third term,

$$\sum_{j=t_i(k)}^{k-1} \mathbb{E}\|\hat{\mathbf{x}}^{j+1} - \hat{\mathbf{x}}^j\|^2 \leq \sum_{j=t_i(k)}^{k-1} \left(3\mathbb{E}\|d^j\|^2 + 3\mathbb{E}\|d^{j+1}\|^2 + 3\mathbb{E}\|\Delta^j\|^2\right)$$

$$\stackrel{equation\ 14}{\leq} 3\sum_{j=t_i(k)}^{k-1} \left(D\sum_{l=(j-D)^+}^{j-1} (\mathbb{E}\|\Delta^l\|^2 + \mathbb{E}\|\Delta^{l+1}\|^2) + \mathbb{E}\|\Delta^j\|^2\right)$$

$$\leq 3\sum_{j=t_i(k)}^{k-1} \left(D\sum_{l=(j-D)^+}^{j-1} (\alpha^2 \sum_{m=(l-D)^+}^{l-1} (2\sigma^2 + \mathbb{E}\|\nabla_{i_m}\mathbf{f}(\hat{\mathbf{x}}^m)\|^2 +\right.$$

$$\left. \mathbb{E}\|\nabla_{i_{m+1}}\mathbf{f}(\hat{\mathbf{x}}^{m+1})\|^2)) + \alpha^2 \sum_{l=(j-D)^+}^{j-1} (\sigma^2 + \mathbb{E}\|\nabla_{i_l}\mathbf{f}(\hat{\mathbf{x}}^l)\|^2)\right) \tag{15}$$

By equation 14 and the following inequality,

$$\sum_{k=0}^{K-1} \sum_{j=(k-D)^+}^{k-1} a_j \leq D\sum_{k=0}^{K-1} a_k,$$

we have

$$\sum_{k=0}^{K-1} \mathbb{E}\|d^k\|^2 \leq D^2\alpha^2 \left(\sum_{k=0}^{K-1} \mathbb{E}\|\nabla_{i_k}\mathbf{f}(\hat{\mathbf{x}}^k)\|^2 + K\sigma^2\right). \tag{16}$$

Similarly, from equation 15,

$$\sum_{k=0}^{K-1} \sum_{j=t_i(k)}^{k-1} \mathbb{E}\|\hat{\mathbf{x}}^{j+1} - \hat{\mathbf{x}}^j\|^2 \leq 3B\Big(2D^3\alpha^2(\sum_{k=0}^{K-1} \mathbb{E}\|\nabla_{i_k}\mathbf{f}(\hat{\mathbf{x}}^k)\|^2 + K\sigma^2)$$

$$+ D\alpha^2(\sum_{k=0}^{K-1} \mathbb{E}\|\nabla_{i_k}\mathbf{f}(\hat{\mathbf{x}}^k)\|^2 + K\sigma^2)\Big)$$

$$= 3B(D + 2D^3)\alpha^2 \left(\sum_{k=0}^{K-1} \mathbb{E}\|\nabla_{i_k}\mathbf{f}(\hat{\mathbf{x}}^k)\|^2 + K\sigma^2\right) \tag{17}$$

Moreover, since each agent updates at least once every $B$ steps, we have

$$\sum_{k=0}^{K-1} \|\nabla_i\mathbf{f}(\hat{\mathbf{x}}^{t_i(k)})\|^2 \leq B\sum_{k=0}^{K-1} \|\nabla_{i_k}\mathbf{f}(\hat{\mathbf{x}}^k)\|^2. \tag{18}$$

Combining equation 13 and equation 16 to equation 18,

$$\frac{\sum_{k=0}^{K-1} \mathbb{E}\|\nabla\mathbf{f}(\mathbf{x}^k)\|^2}{K} = \frac{\sum_{k=0}^{K-1} \sum_{i=1}^{n} \mathbb{E}\|\nabla_i\mathbf{f}(\mathbf{x}^k)\|^2}{K}$$

$$\leq 3n\left(\frac{\sum_{k=0}^{K-1} \mathbb{E}\|\nabla_i\mathbf{f}(\hat{\mathbf{x}}^{t_i(k)})\|^2}{K} + C_0L^2\alpha^2(\frac{\sum_{k=0}^{K-1} \mathbb{E}\|\nabla_{i_k}\mathbf{f}(\hat{\mathbf{x}}^k)\|^2}{K} + \sigma^2)\right)$$

$$\leq 3n\left((B + C_0L^2\alpha^2)\frac{\sum_{k=0}^{K-1} \mathbb{E}\|\nabla_{i_k}\mathbf{f}(\hat{\mathbf{x}}^k)\|^2}{K} + C_0L^2\alpha^2\sigma^2\right),$$

where $C_0 = D^2 + 3B^2(D + 2D^3)$.

By Lemma C.3,

$$\frac{\sum_{k=0}^{K-1} E\|\nabla\mathbf{f}(\mathbf{x}^k)\|^2}{K} \leq \frac{3n(B + C_0L^2\alpha^2)}{\alpha(1 - (D + \frac{1}{2})L\alpha)} \frac{\mathbf{f}(\mathbf{x}^0) - \mathbf{f}^*}{K} + \alpha\left(3nC_0L^2\alpha + \frac{L(D + 1)}{2(1 - (D + \frac{1}{2})L\alpha)}\right)\sigma^2.$$

## C.2 A COROLLARY ON LEMMA C.3

We provide a corollary on Lemma C.3, specifying its convergence rate under a specific step size.

**Corollary C.4.** *For problem 2, given Assumption 3.1 - 3.3, when $\alpha = \frac{1}{2(D+1/2)L\sqrt{K}}$, the sequence $\{\mathbf{x}^k\}$ generated by (4) satisfies the following relations:*

$$\frac{\sum_{k=0}^{K-1} \mathbb{E}\|\nabla \mathbf{f}(\mathbf{x}^k)\|^2}{K} \leq nLC_1 \frac{\mathbf{f}(\mathbf{x}^0) - \mathbf{f}^*}{\sqrt{K}} + \left(\frac{1}{\sqrt{K}} + \frac{3nC_2}{4K}\right)\sigma^2,$$

*where $C_1 = 6B^2D^2 + 3B^2 + 3BD + 3B + D$ and $C_2 = 6B^2D + 3B^2/D + 1$.*

*Proof.* When $\alpha = \frac{1}{2(D+1/2)L\sqrt{K}}$, we have $L\alpha \leq \frac{1}{2(D+1/2)}$ and $1 - (D+1/2)L\alpha \leq 1/2$.

Thus the first term on the RHS of Lemma 3.8 can be upper bounded by

$$3n\left(B + \frac{C_0}{3(D+1/2)^2}\right)(D+1/2)L\frac{\mathbf{f}(\mathbf{x}^0) - \mathbf{f}^*}{\sqrt{K}}. \tag{19}$$

Likewise, its second term can be upper bounded by

$$\left(\frac{3nC_0}{4(D+1/2)^2K} + \frac{L(D+1)}{2(D+1/2)L\sqrt{K}}\right)\sigma^2. \tag{20}$$

Substituting (19) and (20) to Lemma 3.8, we have

$$\frac{\sum_{k=0}^{K-1} \mathbb{E}\|\nabla \mathbf{f}(\mathbf{x}^k)\|^2}{K} \leq 3n\left(B + \frac{C_0}{3(D+1/2)^2}\right)(D+\frac{1}{2})L\frac{\mathbf{f}(\mathbf{x}^0) - \mathbf{f}^*}{\sqrt{K}} + \left(\frac{3nC_0}{4(D+1/2)^2K} + \frac{L(D+1)}{2(D+1/2)L\sqrt{K}}\right)\sigma^2$$

$$\leq 3nL\left(B(D+1) + \frac{C_0}{3D}\right)\frac{\mathbf{f}(\mathbf{x}^0) - \mathbf{f}^*}{\sqrt{K}} + \left(\frac{3nC_0}{4D^2K} + \frac{1}{\sqrt{K}}\right)\sigma^2$$

$$\leq nLC_1 \frac{\mathbf{f}(\mathbf{x}^0) - \mathbf{f}^*}{\sqrt{K}} + \left(\frac{1}{\sqrt{K}} + \frac{3nC_2}{4K}\right)\sigma^2, \tag{21}$$

where $C_1 = 6B^2D^2 + 3B^2 + 3BD + 3B + D$ and $C_2 = 6B^2D + 3B^2/D + 1$. $\qquad\square$

## C.3 PROOF OF THEOREM 3.9

As mentioned, ADSGD with double step size $\{\alpha, \beta\}$ on problem 1 can be viewed as ABCD with step size $\beta$ on the function $L_\alpha(\mathbf{x}) = F(\mathbf{x}) + \frac{\mathbf{x}^T(I-\mathbf{W})\mathbf{x}}{2\alpha}$, where $F(\mathbf{x}) = \sum_{i=1}^n f_i(x_i)$.

W.L.O.G., we assume identical initialization. Note that $L_\alpha(\mathbf{x})$ is $L_L$-smooth. By Lemma 3.8, when $\beta < \frac{1}{(D+\frac{1}{2})L_L}$,

$$\frac{\sum_{k=0}^{K-1} E\|\nabla L_\alpha(\mathbf{x}^k)\|^2}{K} \leq \frac{3n(B + C_0L_L^2\beta^2)}{\beta(1 - (D+\frac{1}{2})L_L\beta)}\frac{\sum_{i=1}^n(f_i(x^0) - f_i^*)}{K}$$

$$+ \beta\left(3nC_0L_L^2\beta + \frac{L_L(D+1)}{2(1-(D+\frac{1}{2})L_L\beta)}\right)\sigma^2, \tag{22}$$

We now bound $\mathbb{E}\|\nabla f(\bar{x}^k)\|^2$ by the relations between $f, F$, and $L_\alpha$.

$$\mathbb{E}\|\nabla f(\bar{x}^k)\|^2 = \mathbb{E}\|\sum_{i=1}^n \nabla f_i(\bar{x}^k)\|^2$$

$$\leq 2\mathbb{E}\|\sum_{i=1}^n \nabla f_i(x_i^k)\|^2 + 2\mathbb{E}\|\sum_{i=1}^n (\nabla f_i(\bar{x}^k) - \nabla f_i(x_i^k))\|^2$$

$$\leq 2\mathbb{E}\|\sum_{i=1}^n \nabla f_i(x_i^k)\|^2 + 2n(\max_i L_i)^2 \mathbb{E}\|1_n \otimes \bar{x}^k - \mathbf{x}^k\|^2 \tag{23}$$

For the first term,

$$\mathbb{E}\|\sum_{i=1}^{n}\nabla f_i(x_i^k)\|^2 = \mathbb{E}\|\sum_{i=1}^{n}\nabla_i F(\mathbf{x}^k)\|^2$$

$$= \mathbb{E}\|\sum_{i=1}^{n}(\nabla_i L_\alpha(\mathbf{x}^k) - \sum_{j=1}^{n}[I - \mathbf{W}]_{ij}x_j^k)\|^2$$

$$= \mathbb{E}\|\sum_{i=1}^{n}\nabla_i L_\alpha(\mathbf{x}^k)]\|^2$$

$$\leq n\mathbb{E}\|\nabla L_\alpha(\mathbf{x}^k)\|^2, \tag{24}$$

where the third equality is from the doubly stochasticity of $\mathbf{W}$.

For the second term, since $1_n \otimes \bar{x}^k - \mathbf{x}^k$ is in the range space of $I - \mathbf{W}$,

$$\mathbb{E}\|1_n \otimes \bar{x}^k - \mathbf{x}^k\|^2 \leq \mathbb{E}\left[\frac{(\mathbf{x}^k)^T(I - \mathbf{W})\mathbf{x}^k}{\lambda_{\min}(I - \mathbf{W})}\right]$$

$$= \frac{2\alpha}{1 - \lambda_2(W)}\mathbb{E}\left[L_\alpha(\mathbf{x}^k) - F(\mathbf{x}^k)\right]$$

$$\leq \frac{2\alpha}{1 - \lambda_2(W)}\mathbb{E}\left[\xi_{L_\alpha}^k - F^*\right]$$

$$\overset{equation\ 12}{\leq} \frac{2\alpha}{1 - \lambda_2(W)}\left(\sum_{i=1}^{n}(f_i(x^0) - f_i^*) + \frac{D+1}{2}KL_L\beta^2\sigma^2\right), \tag{25}$$

where $\lambda_{\min}(\cdot)$ is the minimal positive eigenvalue.

Bringing equation 25 and equation 24 back to equation 23,

$$\mathbb{E}\|\nabla f(\bar{x}^k)\|^2 \leq 2n\mathbb{E}\|\nabla L_\alpha(\mathbf{x})\|^2 + \frac{4n(\max_i L_i)^2\alpha}{1 - \lambda_2(W)}\left(\sum_{i=1}^{n}(f_i(x^0) - f_i^*) + \frac{D+1}{2}KL_L\beta^2\sigma^2\right)$$

Sum the above over $k$ and combine with equation 22,

$$\frac{\sum_{k=0}^{K-1}E\|\nabla f(\bar{x}^k)\|^2}{K} \leq \frac{6n^2(B + C_0 L_L^2\beta^2)}{\beta(1 - (D + \frac{1}{2})L_L\beta)}\frac{\sum_{i=1}^{n}(f_i(x^0) - f_i^*)}{K}$$

$$+ L_L\beta\left(6n^2 C_0 L_L\beta + \frac{n(D+1)}{1 - (D + \frac{1}{2})L_L\beta}\right)\sigma^2$$

$$+ \frac{4n(\max_i L_i)^2\alpha}{1 - \lambda_2(W)}\left(\sum_{i=1}^{n}(f_i(x^0) - f_i^*) + \frac{D+1}{2}KL_L\beta^2\sigma^2\right).$$

## C.4 PROOF OF COROLLARY 3.12

When $\alpha = \frac{2}{L_F K^{1/3}}, \beta = \frac{1}{4L_F(D+1/2)K^{2/3}}$, we have

$$L_L\beta \leq (L_F + \frac{2}{\alpha})\beta$$

$$\leq \frac{1}{4(D+1/2)K^{2/3}} + \frac{1}{8(D+1/2)K^{1/3}}$$

$$\leq \frac{1}{2(D+1/2)K^{1/3}}$$

Thus, the first term on the RHS of Theorem 3.9 can be upper bounded by

$$48n^2\left(B + \frac{C_0}{4(D+1/2)^2}\right)(D+1/2)L_F\frac{\sum_{i=1}^{n}(f_i(x^0) - f_i^*)}{K^{1/3}}. \tag{26}$$

Similarly, its second and third terms can be respectively upper bounded by

$$\left( \frac{3n^2 C_0}{2(D+1/2)K^{2/3}} + \frac{n(D+1)}{(D+1/2)K^{1/3}} \right) \sigma^2 \tag{27}$$

and

$$\frac{8n(\max_i L_i)^2}{(1-\lambda_2(W))L_F K^{1/3}} \left( \sum_{i=1}^{n}(f_i(x^0)) - f_i^* + \frac{D+1}{16(D+1/2)^2 L_F} \sigma^2 \right). \tag{28}$$

Taking equation 26, equation 27, and equation 28 into Theorem 3.9, we have

$$\frac{\sum_{k=0}^{K-1} \mathbb{E}\|\nabla f(\bar{x}^k)\|^2}{K} \leq 48n^2 \left( B + \frac{C_0}{4(D+1/2)^2} \right)(D+1/2)L_F \frac{\sum_{i=1}^{n}(f_i(x^0) - f_i^*)}{K^{1/3}}$$

$$+ \left( \frac{3n^2 C_0}{2(D+1/2)K^{2/3}} + \frac{n(D+1)}{(D+1/2)K^{1/3}} \right) \sigma^2$$

$$+ \frac{8n(\max_i L_i)^2}{(1-\lambda_2(W))L_F K^{1/3}} \left( \sum_{i=1}^{n}(f_i(x^0)) - f_i^* + \frac{D+1}{16(D+1/2)^2 L_F} \sigma^2 \right)$$

$$\leq \left( 48n^2(B(D+1) + \frac{C_0}{3D}) + \frac{8n}{1-\lambda_2(W)} \right) L_F \frac{\sum_{i=1}^{n}(f_i(x^0) - f_i^*)}{K^{1/3}}$$

$$+ \left( \frac{n}{D(1-\lambda_2(W))} + 2n \right) \frac{\sigma^2}{K^{1/3}} + \frac{3n^2 C_0}{2D} \frac{\sigma^2}{K^{2/3}}$$

$$\leq \left( 16n^2 C_1 + \frac{8n}{1-\lambda_2(W)} \right) L_F \frac{\sum_{i=1}^{n}(f_i(x^0) - f_i^*)}{K^{1/3}}$$

$$+ \left( \frac{n}{D(1-\lambda_2(W))} + 2n \right) \frac{\sigma^2}{K^{1/3}} + \frac{3n^2 C_0}{2D} \frac{\sigma^2}{K^{2/3}},$$

where $C_1$ is defined in equation 21.

## D    LLM USAGE

We made limited use of a large language model (LLM) to assist with minor text polishing. Specifically, the LLM was employed to improve grammar, clarity, and readability of certain sentences. All technical content, mathematical derivations, experimental design, and analysis were conceived, written, and verified entirely by the authors. The LLM did not contribute to the generation of ideas, proofs, algorithms, or experimental results.

