# OpenReview forum: "Asynchronous Decentralized SGD for Non-Convex Optimization via a Block-Coordinate Descent Lens"
_ICLR.cc/2026/Conference — ICLR 2026 Conference Withdrawn Submission_

### Official Review · Reviewer_GBmj · 2025-10-16

**Soundness:** 1
**Presentation:** 1
**Contribution:** 1
**Rating:** 2
**Confidence:** 5

**Summary:**

The paper studies asynchronous decentralized optimization. The authors proposed a theoretical tool called Asynchronous Stochastic Block Coordinate Descent (ASBCD), and then developed the convergence analysis of a refined model of Asynchronous Decentralized Stochastic Gradient Descent (ADSGD). Notably, they claimed that the convergence can be proven without any assumption on the bounded data heterogeneity. They also present numerical experiments for ADSGD on different datasets.

**Strengths:**

The paper investigates an important problem of asynchronous decentralized optimization, and studies the convergence rate of a fundamental algorithm called ADSGD. Notably, they claimed that the convergence analysis does not require any assumption on bounded data heterogeneity.

**Weaknesses:**

The convergence rates developed in the paper are very slow! The authors assume that the number of steps between updates is bounded by $B$. Therefore, at the very least, the complexity of the method they proposed should be no worse than the synchronous version multiplied by $B$ (since one can always simulate the synchronous version between any $B$ steps of the asynchronous framework).

However, the main theory in this paper (Theorem 3.9) shows that the optimization term of their method is $n$-times worse than the synchronous complexity multiplied by $B$. Regarding the synchronous complexity, one can for instance check the following gradient tracking paper:
Koloskova, A., Lin, T., and Stich, S. U. An improved analysis of gradient tracking for decentralized machine learning. In Advances in Neural Information Processing Systems, 2021.

Just to ensure I understand their theories correctly, I also looked at the proof in the Appendix. The derivations of many upper bounds are quite loose. I believe that the analysis should be significantly improved before this work is ready for publication.

**Questions:**

The experiment plots also look a bit weird. Could you explain why in Figure 3(b) the ADSGD is even faster than SGD?

---

### Official Review · Reviewer_zpCu · 2025-10-20

**Soundness:** 3
**Presentation:** 3
**Contribution:** 2
**Rating:** 4
**Confidence:** 4

**Summary:**

The paper proposes ADSGD, an Asynchronous Decentralized Stochastic Gradient Descent algorithm for non-convex optimization under bounded computation and communication delays. The main technical idea is to analyze ADSGD through a block-coordinate descent perspective. The authors develop a double-step-size technique that decouples the step size from delay-dependent Lipschitz constants, allowing delay-independent convergence guarantees. They prove convergence to stationary points for ADSGD under bounded computation and communication delays without assuming bounded data heterogeneity.
The analysis establishes an $O(K^{-1/3})$ convergence rate with step sizes independent of computation delays.
Empirically, ADSGD (logistic regression on MNIST; VGG11 on CIFAR-10) is robust to stragglers and shows faster wall-clock convergence than several baselines (ADPSGD, RFAST, DSGD, parallel SGD) across heterogeneous delay patterns and up to 128 agents.

**Strengths:**

- The work extends asynchronous block-coordinate descent theory to the stochastic.

- The analysis does not rely on bounded data heterogeneity assumption and decouples computation delays from step-size constraints.

- The double-step-size technique is interesting.

- Experiments cover heterogeneous data, different delay patterns, and scalability up to 128 agents. Results show performance gains.

- The paper is generally quite well-written.

**Weaknesses:**

1. The theoretical convergence rate is slower than that of standard SGD ($O(K^{-1/3})$ vs $O(K^{-1/2})$).

2. Poor literature review

    - The literature review is outdated. It includes only 1 citation from 2021, 3 from 2022, 4 from 2023, and none from 2024 or 2025. The past two years have seen a lot progress in asynchronous and/or decentralized optimization, yet these developments are entirely absent from the discussion. This omission weakens the positioning of the paper and gives the misleading impression that the proposed results are more novel than they actually are.

    - line 57: The authors claim that "most of the asynchronous algorithms make probabilistic assumptions regarding update patterns". This statement is not very accurate and stems from a selective review of the literature. For instance, the recent works [1, 2, 3, 4] do not make any assumptions about update patterns, unlike this paper, which does impose such assumptions. These works should be acknowledged.

    - line 70: The authors claim that the works on ABCD methods "have not considered the stochastic gradient setting". This is again incorrect. There exist studies, e.g., [5], which analyze stochastic asynchronous block-coordinate descent with variance reduction. The paper should cite and position itself relative to these earlier contributions.

    - The authors of [6] (also not cited) study asynchronous decentralized SGD and obtain convergence guarantees matching those of standard SGD (up to a logarithmic factor), whereas the present work achieves a slower rate. The paper should explain this discrepancy and clarify what lead to the weaker theoretical guarantees. Similarly, [7] is another relevant and recent example that should be acknowledged.

3. Missing or inconsistent notation, for example

    - $\mathcal{N}_i$ in line 127

    - $i_k$ is used inconsistently: in eq. (4) it denotes the active block (scalar) but later the text uses notation $\{i_k\} = [n]$.

4. [1] proposes nearly optimal decentralized stochastic asynchronous optimization methods. These should be included both in theoretical and empirical comparisons.

[1] Tyurin A, Richtárik P. On the optimal time complexities in decentralized stochastic asynchronous optimization. Advances in Neural Information Processing Systems. 2024 Dec 16;37:122652-705.

[2] Tyurin A, Gruntkowska K, Richtárik P. Freya page: First optimal time complexity for large-scale nonconvex finite-sum optimization with heterogeneous asynchronous computations. Advances in Neural Information Processing Systems. 2024 Dec 16;37:54239-87.

[3] Tyurin A, Richtárik P. Optimal time complexities of parallel stochastic optimization methods under a fixed computation model. Advances in Neural Information Processing Systems. 2023 Dec 15;36:16515-77.

[4] Maranjyan A, Tyurin A, Richtárik P. Ringmaster ASGD: The first Asynchronous SGD with optimal time complexity. arXiv preprint arXiv:2501.16168. 2025 Jan 27.

[5] Gu B, Huo Z, Huang H. Asynchronous stochastic block coordinate descent with variance reduction. arXiv preprint arXiv:1610.09447. 2016 Oct 29.

[6] Attiya H, Schiller N. Asynchronous fully-decentralized SGD in the cluster-based model. Theoretical Computer Science. 2025 Mar 21;1031:115073.

[7] Nadiradze G, Sabour A, Davies P, Li S, Alistarh D. Asynchronous decentralized sgd with quantized and local updates. Advances in Neural Information Processing Systems. 2021 Dec 6;34:6829-42.

**Questions:**

1. How does your method and analysis compare to [6, 7]? Why do you achieve worse convergence guarantees?

2. Could the method benefit from variance reduction?

3. line 98: "ADSGD reduces per-iteration communication overhead by 50\% and memory usage by 70\%". Could you clarify the reference point?

4. Could you respond to the issues raised in previous sections?

---

### Official Review · Reviewer_Lzeh · 2025-10-27

**Soundness:** 2
**Presentation:** 2
**Contribution:** 2
**Rating:** 4
**Confidence:** 4

**Summary:**

The paper presents a theoretical analysis of asynchronous decentralized SGD (ADSGD) from a block-coordinate descent perspective. Convergence properties are examined via a double-step-size technique, yielding guarantees that do not rely on bounded data heterogeneity.

**Strengths:**

The paper presents a unified theoretical perspective linking asynchronous decentralized learning and block coordinate descent.
The experimental evaluation covers data heterogeneity and scalability under different cases.

**Weaknesses:**

The paper lacks the inclusion of recent references.
The paper does not provide a scalability comparison with other methods.

**Questions:**

The paper analyzes asynchronous decentralized SGD (ADSGD) from the perspective of block-coordinate descent, introducing a double-step-size technique to establish convergence guarantees under bounded delays.There are several questions as follows.
1.The motivation for the key theoretical connection between ADSGD and ASBCD is unclear.The construction of the augmented function $L_{\alpha}(x)$ seems to be presented as an a posteriori justification rather than a naturally motivated insight. Please explain the conceptual pathway that led to identifying this connection.

2.The description of Algorithm 2 (ADSGD) is confusing regarding the timing of communication and computation. Line 8 suggests that each node keeps receiving models until its gradient is ready, which seems to imply model reception happens before gradient computation. However, the text later claims that ADSGD uses the global model after gradient estimation. Please clarify whether communication and computation occur sequentially or in parallel in Algorithm 2, as the current phrasing (“until $g^{f_i}(x_i)$ is available”) makes this unclear.

3.Figure 4 only shows that ADSGD's performance improves with more agents, but it lacks a critical comparison with baseline methods (e.g., DSGD, ADPSGD) under the same scaling conditions.

---

### Official Review · Reviewer_EcFE · 2025-10-30

**Soundness:** 1
**Presentation:** 1
**Contribution:** 2
**Rating:** 2
**Confidence:** 4

**Summary:**

The paper introduces an asynchronous decentralized SGD method. The authors analyze their approach for non-convex smooth functions under a bounded delay assumption.
The main novelty is that they view their method as a special case of block-coordinate descent and perform the analysis within that framework.
They also show that their method outperforms the baseline approaches that they consider.

**Strengths:**

The main strength of the paper is the analysis from the perspective of block-coordinate descent. I am not sure whether this viewpoint has been explored before (the authors should clarify this in the paper), but if not, it represents an interesting and novel way of analyzing asynchronous decentralized SGD. However, it is not entirely clear whether this perspective provides any concrete advantages over previous approaches (see Weakness 1).

**Weaknesses:**

1. **Unclear theoretical comparison to prior work**
It is not clear whether the proposed method achieves a better theoretical iteration complexity than previous methods, particularly those in [1, 2].
Even more concerning, in [1, Corollary 1] the iteration complexity is $\mathcal{O}\left(\tfrac{1}{K} + \tfrac{\sigma}{\sqrt{K}}\right)$, which is better than the $\mathcal{O}\left(\tfrac{1}{K^{1/3}} + \tfrac{\sigma}{K^{1/3}}\right)$ rate reported in Corollary 3.12 of this paper.
Are the gains mainly in terms of memory or communication efficiency, as suggested in Table 2? The reported improvement (around a 4× reduction) appears relatively minor, and if this comes at the cost of requiring more iterations, the overall benefit is questionable.

2. **Delay-independent stepsize**
The authors claim that one advantage of their method is that the stepsize does not depend on the delay. However, the paper does not explain why this is beneficial; in fact, this may be a disadvantage.
In [3, 4], delay-adaptive stepsizes lead to better iteration complexity and improved dependence on delay.
Similarly, the optimal asynchronous methods in the centralized setting [5, 6] explicitly use the delay bound in the stepsize.
Intuitively, scaling the stepsize with the delay makes sense, since in the stochastic setting the delay bound acts similarly to an effective batch size [5].

3. **Dependence on maximum delay**
The convergence rate depends on the maximum delay (and even its square), which is undesirable.
If a single iteration experiences a large delay while others are small, this should not dominate the overall complexity.
Related works in the centralized setting [3, 4] have addressed this issue using delay-adaptive stepsizes, which lead to improved dependence on the average delay rather than the worst-case delay.

4. **No time complexity analysis**
The paper analyzes only iteration complexity, which is insufficient for evaluating efficiency in asynchronous optimization.
Iteration complexity alone does not reveal whether the method is actually faster than synchronous counterparts.
In fact, asynchronous methods often have worse iteration complexity, and to demonstrate their advantage, one must compare them in terms of time complexity instead.
It is also not guaranteed that asynchronous methods achieve better time complexity—sometimes they match the synchronous ones, as shown in Table 1 of [6].
In such cases, these asynchronous methods are not desirable, since they achieve the same time complexity while performing more computations.
Time-complexity analyses for asynchronous methods can be found in [5, 6, 7] for the centralized case and in [8] for the decentralized setting.

5. **Missing comparison to state-of-the-art**
There exists an optimal method in the same non-convex smooth decentralized regime, Amelie SGD [8], which achieves optimal time complexity guarantees.
The paper does not include any theoretical or empirical comparison with this method.

I understand that the authors may be unaware of the recent line of work on time-complexity analysis, so it might be acceptable to overlook Weaknesses 4 and 5 to some extent.
However, they should at least cite this line of research and provide a numerical comparison with the state-of-the-art method, since a thorough theoretical comparison may take additional effort.

---

[1] Zehan Zhu, Ye Tian, Yan Huang, Jinming Xu, and Shibo He. "Robust fully-asynchronous methods for distributed training over general architecture". arXiv preprint arXiv:2307.11617, 2023.

[2] Vyacheslav Kungurtsev, Mahdi Morafah, Tara Javidi, and Gesualdo Scutari. "Decentralized asynchronous non-convex stochastic optimization on directed graphs". IEEE Transactions on Control of Network Systems, 2023.

[3] Anastasiia Koloskova, Sebastian U Stich, and Martin Jaggi. "Sharper convergence guarantees for asynchronous SGD for distributed and federated learning". Advances in Neural Information Processing Systems, 35:17202–17215, 2022.

[4] Konstantin Mishchenko, Francis Bach, Mathieu Even, and Blake E Woodworth. "Asynchronous SGD beats minibatch SGD under arbitrary delays". Advances in Neural Information Processing Systems, 35:420–433, 2022.

[5] Artavazd Maranjyan, Alexander Tyurin, and Peter Richt´arik. "Ringmaster ASGD: The first Asynchronous SGD with optimal time complexity". In the International Conference on Machine Learning, 2025

[6] Artavazd Maranjyan and Peter Richt´arik. "Ringleader ASGD: The first Asynchronous SGD with optimal time complexity under data heterogeneity". arXiv preprint arXiv:2509.22860, 2025

[7] Alexander Tyurin and Peter Richt ´arik. "Optimal time complexities of parallel stochastic optimization methods under a fixed computation model. Advances in Neural Information Processing Systems, 36, 2024.

[8] Alexander Tyurin and Peter Richtárik. "On the optimal time complexities in decentralized stochastic asynchronous optimization". Advances in Neural Information Processing Systems 37 (2024): 122652-122705.

**Questions:**

1. Can you provide a proper comparison with previous methods in terms of iteration complexity? Is your method theoretically better in that regard?
2. Other than memory and communication reduction, what are the main theoretical benefits of your approach compared to existing methods?
3. In Corollary C.4, the convergence rate appears worse, at $\mathcal{O}(1/\sqrt{K})$. Could the stepsize $\gamma$ be chosen to achieve a faster $\mathcal{O}(1/K)$ rate instead?
4. Could you include a comparison with Amelie SGD [8], at least in the experimental section, to better position your method relative to the state of the art?

---

### Note · Authors · 2025-12-14

I have read and agree with the venue's withdrawal policy on behalf of myself and my co-authors.